



# Cloud processing of DMS oxidation products limits SO₂ and OCS production in the Eastern North Atlantic marine boundary layer

Delaney B. Kilgour[1], Christopher M. Jernigan[1,*,#], Olga Garmash[2,^], Sneha Aggarwal[3,4], Claudia Mohr[3,†,+], Matt E. Salter[3,4], Joel A. Thornton[2], Jian Wang[5], Paul Zieger[3,4], Timothy H. Bertram[1]

[1]Department of Chemistry, University of Wisconsin-Madison, Madison, WI 53706, USA
[2]Department of Atmospheric Sciences, University of Washington, Seattle, WA 98195, USA
[3] Department of Environmental Science, Stockholm University, Stockholm, 10691, Sweden
[4]Bolin Centre for Climate Research, Stockholm University, Stockholm, 10691, Sweden
[5]Center for Aerosol Science and Engineering, Department of Energy, Environmental and Chemical Engineering, Washington University in St. Louis, St. Louis, MO 63130, USA
[*]Now at Cooperative Institute for Research in Environmental Sciences, University of Colorado, Boulder, CO 80305, USA
[#]Now at NOAA Chemical Sciences Laboratory, Boulder, CO 80305, USA
[^]Now at Department of Chemistry, University of Copenhagen, DK-2100 Copenhagen Ø, Denmark
[†]Now at Department of Environmental Systems Science, ETH Zurich, 8092 Zürich, Switzerland
[+]Now at Laboratory of Atmospheric Chemistry, Paul Scherrer Institute, 5232 Villigen, Switzerland

*Correspondence to*: Timothy H. Bertram (timothy.bertram@wisc.edu)

**Abstract.** Dimethyl sulfide (DMS) is the major sulfur species emitted from the ocean. The gas-phase oxidation of DMS by hydroxyl radicals proceeds through the stable, soluble intermediate hydroperoxymethyl thioformate (HPMTF), eventually forming carbonyl sulfide (OCS) and sulfur dioxide (SO₂). Recent work has shown that HPMTF is efficiently lost to marine boundary layer (MBL) clouds, thus arresting OCS and SO₂ production and their contributions to new particle formation and growth events. To date, no long-term field studies exist to assess the extent to which frequent cloud processing impacts the fate of HPMTF. Here we present six weeks of measurements of cloud fraction and the marine sulfur species, methanethiol, DMS, and HPMTF, made at the ARM Research Facility on Graciosa Island, Azores, Portugal. Using an observationally constrained chemical box model, we determine that cloud loss is the dominant sink of HPMTF in this region of the MBL during the study, accounting for 79-91% of HPMTF loss on average. When accounting for HPMTF uptake to clouds, we calculate a campaign average reduction in DMS-derived MBL SO₂ and OCS of 52-60% and 80-92% for the study period. Using yearly measurements of site- and satellite-measured 3-dimensional cloud fraction and DMS climatology, we infer that HPMTF cloud loss is the dominant sink of HPMTF in the Eastern North Atlantic during all seasons, and occurs on timescales faster than what is prescribed in global chemical transport models. Accurately resolving this rapid loss of HPMTF to cloud has important implications for constraining drivers of MBL new particle formation.

## 1 Introduction

Aqueous reactions in clouds can significantly alter trace gas and aerosol budgets by acting as efficient, terminal sinks for water-soluble species and sites for the formation of reactive products (Barth et al., 2021; Li et al., 2017; Yang et al., 2015).



Examples include cloud scavenging of dinitrogen pentoxide ($N_2O_5$) and nitric acid ($HNO_3$) influencing the tropospheric $NO_x$
budget (Holmes et al., 2019; Levine and Schwartz, 1982), uptake of sulfur dioxide ($SO_2$) contributing to aerosol production
and acid rain (Irwin and Williams, 1988), cloud processing increasing the production of isoprene secondary organic aerosol
(SOA) (Lamkaddam et al., 2021), and cloud chemistry controlling the conversion of dimethyl sulfide ($CH_3SCH_3$; DMS) to
$SO_2$ and methane sulfonic acid ($CH_3SO_3H$; MSA) (Chen et al., 2018; Hoffmann et al., 2016). By redistributing chemical
budgets in the lower troposphere, cloud processing can consequently affect the spatial distribution and availability of vapors
to contribute to new particle formation (Novak et al., 2021), the concentration of cloud condensation nuclei (CCN) (Feingold
et al., 1998), and the magnitude of long-lived climate forcing products (Jernigan et al., 2022a; Novak et al., 2021).

For water-soluble species with high uptake coefficients that react irreversibly, uptake to cloud droplets is limited by gas-phase
diffusion to the droplet surface, leading to in-cloud lifetimes for typical cloud conditions on the order of ten seconds or less
(Holmes et al., 2019; Levine and Schwartz, 1982). Large eddy simulation studies indicate the residence time of air within the
cloud is significantly longer, ranging from 15 minutes to two hours for many stratus and stratocumulus clouds (Feingold et al.,
1998, 2013; Kogan, 2006; Stevens et al., 1996), and even longer for cirrus clouds (Podglajen et al., 2016). This results in the
complete and rapid removal of water-soluble molecules that react irreversibly in the cloud layer. As such, cloud processing of
water-soluble species with irreversible uptake in the well-mixed boundary layer is dependent on the mixing rate of clear air
into cloud, here referred to as the cloud entrainment rate (Holmes et al., 2019). A new method using entrainment-limited
uptake, incorporating grid cell cloud fraction from satellite reanalysis products (MERRA-2) and entrainment into the kinetic
rate expression, was recently developed to account for cloud uptake (Holmes, 2022; Holmes et al., 2019), and has been
implemented into global chemical transport models to evaluate chemical budgets for a variety of species, including halogen,
sulfur, and nitrogen-containing molecules (Alexander et al., 2020; Chan et al., 2021; Holmes et al., 2019; Jernigan et al., 2022a;
Novak et al., 2021; Shah et al., 2021; Wang et al., 2021). This method utilizes an average entrainment rate of 1 h⁻¹ based on
mean values of cloud residence time for stratus and stratocumulus clouds scaled by satellite-derived 3D cloud fraction (Holmes
et al., 2019). This entrainment-limited method has been shown to be more physically accurate and less computationally
expensive than previous parameterizations for cloud uptake, such as the thin cloud (Parrella et al., 2012) and cloud partitioning
(Tost et al., 2006) approximations.


Trace gas uptake by clouds can play a particularly important role in trace gas and aerosol budgets in the marine boundary layer
(MBL) due to the large and persistent cover of low-level clouds over the oceans. Globally, stratus and stratocumulus clouds
are present over 10-70% of the MBL, and their coverage can exceed 50% in the annual mean over subtropical and midlatitude
oceans (Wood, 2012). More recent estimates reported five-year averaged low-level cloud fractions even larger, exceeding 70%
in the subtropics and the extratropical North Atlantic, North Pacific, and Southern Oceans, regions where stratocumulus cloud
decks are common (Naud et al., 2023). Given the ocean is the largest natural source of reduced sulfur to the atmosphere,
primarily in the form of DMS (~27.1 Tg S yr⁻¹) (Andreae, 1990; Bates et al., 1992; Hulswar et al., 2022) and to a lesser extent,



methanethiol ($CH_3SH$; MeSH) (Novak et al., 2022), low-level MBL clouds have the potential to impact the sulfur budget globally through the uptake of their soluble oxidation intermediates.


DMS is formed in the ocean as one of two major degradation products of the precursor algal metabolite dimethylsulfoniopropionate (DMSP) (Challenger and Simpson, 1948). The other DMSP degradation product is MeSH (Kiene, 1996). Once emitted to the atmosphere, the primary fate of both DMS and MeSH is reaction with hydroxyl radicals (OH), with the lifetime of DMS to OH approximately five times longer than that of MeSH to OH at 298 K (Burkholder et al., 2019). The

OH-oxidation of MeSH and subsequent $O_2$ addition forms the $CH_3SO_2$ radical, which has a temperature-dependent branching ratio forming $SO_2$ or MSA (Chen et al., 2023). Recent computational work has shown the $SO_2$ yield from $CH_3SO_2$ is 99% at 300 K, but drops to 4% at 260 K (Chen et al., 2023). The OH-oxidation of DMS is also highly temperature-dependent, proceeding either by OH-addition (~30% at 298 K) or by H-abstraction (~70% at 298 K). The OH-addition pathway leads to the formation of several soluble products, including MSA, methane sulfinic acid ($CH_4O_2S$; MSIA), dimethyl sulfoxide

($CH_3SOCH_3$; DMSO), and dimethyl sulfone ($C_2H_6O_2S$; $DMSO_2$), and primarily contributes to particle growth (Barnes et al., 1994; Conley et al., 2009; Hoffmann et al., 2016). The H-abstraction pathway produces the methylthiomethyl peroxy radical ($CH_3SCH_2OO$; MTMP), which can undergo intramolecular hydrogen shift rearrangements and additions of $O_2$ to form the stable, soluble intermediate hydroperoxymethyl thioformate ($HOOCH_2SCHO$; HPMTF) (Berndt et al., 2019; Wu et al., 2015). This isomerization pathway to HPMTF production competes with bimolecular reactions between MTMP and NO, $HO_2$, and

$RO_2$ (Berndt et al., 2019), which are typically in low concentration in the marine atmosphere (<15 ppt, <15 ppt, and <150 ppt, respectively) (Creasey et al., 2003; Lee et al., 2009; Vaughan et al., 2012). Once formed, HPMTF is further oxidized by OH to carbonyl sulfide (OCS) (Jernigan et al., 2022a) and $SO_2$ (Veres et al., 2020), leading to new sulfate ($SO_4^{2-}$) aerosol particle formation through the production of sulfuric acid ($H_2SO_4$). Recent aircraft measurements found HPMTF was globally ubiquitous in the MBL (Veres et al., 2020) and global chemical transport modeling showed it is the dominant reservoir of

DMS oxidation products; analyses in this study indicated the yield of HPMTF from the DMS H-abstraction pathway ($\alpha_{HPMTF}$) was 0.76, and estimated 46% of all emitted DMS globally formed HPMTF (Novak et al., 2021). HPMTF has also been shown to be efficiently depleted in MBL cloud suggesting irreversible loss (Novak et al., 2021; Siegel et al., 2023; Veres et al., 2020; Vermeuel et al., 2020), which is briefly summarized below.

Aircraft measurements by Veres et al. (2020) first showed the rapid depletion of HPMTF within the MBL cloud layer during ATom 3 and ATom 4, reporting on average a 75% reduction in HPMTF in the presence of cloud. This result was subsequently supported by several qualitative findings at ground sites. For example, in coastal Southern California, Vermeuel et al. (2020) found that observed HPMTF diurnal profiles could only be reproduced by a model when including a time-dependent HPMTF cloud loss based on GOES imagery. In the Arctic, Siegel et al. (2023) measured reduced HPMTF in cloudy and semi-cloudy

conditions compared to cloud-free conditions. The only existing collocated measurements of DMS and HPMTF are from a flight off the coast of Southern California (Novak et al., 2021). In cloud-free conditions, average [DMS]/[HPMTF] was low





(1.25), but was much higher (20) below the cloud deck, evidence for cloud processing of the DMS-oxidation product, HPMTF. Analysis of eddy covariance flux measurements of HPMTF on this same flight produced the only quantified loss rate of HPMTF to cloud currently in the literature (Novak et al., 2021). The timescale of HPMTF loss to a stratocumulus cloud deck was $1.2 \pm 0.6$ h, which was greater than four times faster than other HPMTF loss pathways. This irreversible cloud uptake of HPMTF weakens the links between DMS and the climate forcing products OCS and $SO_2$ along the H-abstraction pathway, and subsequent new particle formation and CCN production. Global model analyses incorporating the HPMTF cloud loss term determined from Novak et al. (2021) indicated cloud chemistry reduced $SO_2$ production from DMS globally by 35% (Novak et al., 2021) and OCS production globally by 92% (Jernigan et al., 2022a). Further, the prompt conversion of aqueous HPMTF in cloud to $SO_4^{2-}$ at unit yield (Jernigan et al., 2024) could significantly increase $SO_4^{2-}$ concentrations while bypassing new particle formation (Novak et al., 2021).

However, to date, no long-term field studies exist with coincident measurements of DMS and HPMTF. This limits our ability to assess how cloud chemistry impacts DMS oxidation on long time scales, where cloud fraction and cloud type are expected to vary. Here we present six weeks of *in situ* measurements of the reactant and product pair DMS and HPMTF, and MeSH, made in the Eastern North Atlantic (ENA). We use these gas-phase measurements and extensive observations of atmospheric and cloud properties made at the Atmospheric Radiation Measurement (ARM) Research Facility on Graciosa Island, Azores, Portugal, with a chemical box model to determine how frequent cloud processing impacts the conversion of DMS to $SO_2$ and OCS in the MBL. We show that over a six-week period, cloud uptake is the dominant loss process of HPMTF and occurs at rates significantly faster than what is currently prescribed in global chemical transport models where uptake is scaled by satellite-derived cloud fraction.

## 2 Methods

### 2.1 Measurements of gas-phase sulfur species at ENA

Continuous, real-time measurements of DMS, MeSH, and HPMTF were made from June 1, 2022 to July 15, 2022 at the ENA ARM Research Facility on Graciosa Island, Azores, Portugal (39.0916 °N, 28.0257 °W, 30 m elevation) as part of the Aerosol Growth in the Eastern North Atlantic (AGENA) project. DMS and MeSH were measured at 10 m above ground level with a Vocus proton transfer reaction time-of-flight mass spectrometer (RT-Vocus; Aerodyne Research, Inc. and Tofwerk AG) (Krechmer et al., 2018). Full details of the RT-Vocus sampling at AGENA and quantifications of DMS and MeSH are reported in Kilgour et al. (2024). Collocated HPMTF measurements at 4 m above ground level were made with a chemical ionization time-of-flight mass spectrometer (Aerodyne Research, Inc. and Tofwerk AG) equipped with a medium-pressure (50 mbar) Vocus AIM reactor (Riva et al., 2024). Multiple reagent ions, namely iodide, bromide, and benzene, were generated using VUV lamps to target a wide range of oxygenated and non-oxygenated compounds. HPMTF was detected with the iodide reagent ion as an adduct ion with iodide ($[I \cdot C_2H_4O_3S]^-$). This mass was <0.1 m/$Q$ from $N_2O_5$, and at the same unit mass as



several other peaks (Supplemental S1). The instrument resolution ($m/\Delta m$ = 5500) did not enable separable peak fitting of the closest two peaks, HPMTF and $N_2O_5$, leading to some early morning interference in HPMTF when $N_2O_5$ was present. However, the $N_2O_5$ signal was small due to low $NO_x$ ambient conditions, and HPMTF and $N_2O_5$ had different diurnal profiles, resulting in minimal impact overall for this analysis. Vocus AIM zeros were completed every hour at the capillary, and HPMTF was quantified post-campaign with the experimentally-determined humidity-dependent calibration factor for formic acid based on the similar iodide adduct binding enthalpies for HPMTF and formic acid (Iyer et al., 2016; Jernigan et al., 2022a). By comparing clear sky measurements of HPMTF to modelled clear sky HPMTF, we estimate HPMTF concentrations reported here, using the formic acid calibration factor, are underestimated by up to 60% due to a combination of inlet loss of HPMTF and lack of an authentic calibration standard (Fig. S1). As a result, all of the following reported measurements of [HPMTF] and [DMS]/[HPMTF], which use the calibration factor to formic acid, should be interpreted as a lower limit and upper limit, respectively. More details on the HPMTF measurement, quantification, and derivation of its uncertainty are in Supplemental S1. Limits of detection for a signal-to-noise ratio of 3 at 5-minute averaging for DMS, MeSH, and HPMTF were 1.8, 5.1, and 0.1 ppt, respectively (Bertram et al., 2011). HPMTF was below the detection limit in 19% of 5-minute averaged HPMTF data points; diurnally, this was largest in the early morning at Hour 7 (32%) and lowest in the afternoon at Hour 16 (3.6%), where these and the following hours are in local time. Points below the detection limit were replaced with half the detection limit for reporting statistics and interpreting [DMS]/[HPMTF] ratios (Antweiler and Taylor, 2008). Since the subsequent analysis utilizes afternoon [DMS]/[HPMTF] ratios, the treatment of the detection limit had a minimal effect relative to all other sources of uncertainty. Lastly, DMS, MeSH, and HPMTF were insensitive to nearby Graciosa airport activity and so no pollution flag was applied to the measurements in this work, contrary to those in Kilgour et al. (2024).

## 2.2 Development of a box model to derive HPMTF cloud loss rates from [DMS]/[HPMTF]

A coupled ocean-atmosphere 0-D chemical box model was created in the Framework for 0-D Atmospheric Modeling (F0AM, Wolfe et al., 2016), implementing the Master Chemical Mechanism (MCM) v3.3.1 (http://mcm.york.ac.uk (last access: 5 Dec 2023) (Jenkin et al., 1997; Saunders et al., 2003), and updated sulfur chemistry for MeSH, HPMTF, and other DMS oxidation products. The constrained box model was used to determine the rate of HPMTF lost to cloud as discussed below.

The box model was run with a four-day spin-up period to allow reactive intermediates to reach equilibrium. Diurnally-averaged measurements from the Aerosol Observing System (AOS) (Uin et al., 2019) and RT-Vocus during the study period were used as inputs to constrain pressure, temperature, humidity, and trace gas concentrations ($O_3$, CO, VOC). The emission flux of DMS ($4.5 \times 10^9$ molec. $cm^{-2}$ $s^{-1}$) was prescribed to match the observed study diurnal-average mixing ratio (diurnal minimum 80 ppt – diurnal maximum 137 ppt) and was within the range of typical oceanic DMS emission fluxes ($0-7.0 \times 10^9$ molec. $cm^{-2}$ $s^{-1}$) (Hulswar et al., 2022). A constant OH profile peaking at $4.5 \times 10^6$ molec. $cm^{-3}$ (diurnal average of $1.3 \times 10^6$ molec. $cm^{-3}$) was used. This OH profile agreed well with previous predictions of the zonally averaged surface OH concentration for July at this latitude of $1.49 \times 10^6$ molec. $cm^{-3}$ (Spivakovsky et al., 2000). This OH concentration from climatological analysis was



determined in a photochemical model constrained by surface and column observations of variables affecting OH, such as the concentrations of $O_3$, water vapor, nitrogen oxides, CO, hydrocarbons, and temperature and cloud optical depth (Spivakovsky et al., 2000). OH production below and above cloud was assumed to be approximately equivalent, based on the following two

pieces of evidence: (1) <15% difference in measured $J(O^1D)$ in cloudy and clear conditions in the North Pacific during ATom (Hall et al., 2018), and (2) given DMS and MeSH are co-emitted species with different, known OH loss rates, [DMS]/[MeSH] can provide insight into OH exposure. Our measurements indicate no dependence in measured midday [DMS]/[MeSH] on cloud fraction (Fig. S2). As a result, the model used a constant OH profile, independent of cloud fraction, to interpret cloud loss of HPMTF across the study. Based on average boundary layer heights determined from sonde profiles approximately

every 12 hours during the study (average $1009 \pm 312$ m ($1\sigma$) and interquartile range $748 – 1240$ m), a static boundary layer height of 1000 m was assumed. Free troposphere – boundary layer mixing was treated as a first order dilution term, calculated using an exchange velocity of 0.5 cm s$^{-1}$ (Faloona, 2009) and a 1000 m boundary layer height. Additional discussion on the dependence of boundary layer height and exchange velocity on the fraction of DMS oxidized in the MBL can be found in Section 3.2.1.

In the model, HPMTF was formed chemically via the temperature-dependent isomerization of MTMP (Assaf et al., 2023) and lost via OH-oxidation (Jernigan et al., 2022a), dry deposition (Vermeuel et al., 2020), aerosol uptake (Jernigan et al., 2022b), and a variable fourth term, interpreted as cloud loss. HPMTF OH-oxidation was set to $1.4 \times 10^{-11}$ cm$^3$ molec.$^{-1}$ s$^{-1}$ forming $SO_2$ at 87% yield and OCS at 13% yield (Jernigan et al., 2022a), HPMTF dry deposition was set to 0.75 cm s$^{-1}$ and was independent

of wind speed over the range of wind speeds observed (Vermeuel et al., 2020), and uptake to marine aerosol particles was calculated according to Eq. 1, where $A$ is the aerosol surface area density, $D_g$ is the diffusivity in air, $r$ is the aerosol radius, $v$ is the mean molecular speed, and $\gamma$ is the reactive uptake coefficient. A constant aerosol surface area of 45.0 μm$^2$ cm$^{-3}$ was used, corresponding to the median dry aerosol surface area measured by a scanning mobility particle sizer (measures 10 nm – 1000 nm diameter particles) during the six week study (20.0 μm$^2$ cm$^{-3}$ and 12.6-25.3 μm$^2$ cm$^{-3}$ interquartile range) with an

estimated hygroscopic growth factor of 1.5 applied (Zhang et al., 2014). The reactive uptake coefficient, $\gamma$, was set to 0.0016, corresponding to an experimentally-measured value for deliquesced NaCl particles (Jernigan et al., 2022b). Aerosol uptake (De Bruyn et al., 1994; Hoffmann et al., 2021) and dry deposition equivalent to the HPMTF dry deposition (Johnson, 2010; Vermeuel et al., 2020) was also included for the DMS oxidation products MSIA, MSA, DMSO, and DMSO$_2$, but the model did not treat cloud loss of these species. A complete table of updated model chemical reactions relevant to DMS, MeSH, and

HPMTF is included in Table S1.

$$k = A \left(\frac{r}{D_g} + \frac{4}{v\gamma}\right)^{-1} \tag{1}$$



The difference between the clear sky modelled [DMS]/[HPMTF] diurnal profile and measurements of [DMS]/[HPMTF]
diurnal profiles during the study were used to assign a fourth term, interpreted as the rate of HPMTF cloud loss. This was
completed for the 31 study days with at least 20% data coverage in Hours 14-15 and at least 25% data coverage in Hours 13-
17. These thresholds were selected to ensure data coverage when the diurnal profile of [DMS]/[HPMTF] was at a stable
minimum. Since measured [DMS]/[HPMTF] is an upper limit, as discussed in Section 2.1, the derived cloud loss rates from
residual [DMS]/[HPMTF] should also be interpreted as an upper limit. This same analysis was completed with [HPMTF]
corrected so clear sky measurements of HPMTF agreed with clear sky modelled HPMTF (Fig. S1). From this analysis, we
estimate cloud loss terms of HPMTF are an overestimate by up to a factor of three. Loss rates of HPMTF to cloud and the
fractional loss of HPMTF to individual pathways below are reported as ranges based on this uncertainty to more accurately
compare to literature values. More details on derivation of cloud loss terms are in Section 3.2.2.

**2.3 Supporting measurements**

Continuous measurements at ENA provided by ARM were used in tandem with DMS, MeSH, and HPMTF measurements to
evaluate trends in HPMTF cloud loss rates. Best estimates of cloud base height (CBH) were determined from ceilometer and
micropulse lidar measurements saved at 1 Hz (Johnson et al., 2022). Boundary layer heights (BLH) for the study period were
determined manually based on inflection points in potential temperature and water mixing ratio (Albrecht et al., 1995) in sonde
measurements launched two to three times per day (Riihimaki et al., 2022). Well-mixed boundary layers had vertical slopes in
both potential temperature and mixing ratio below the inversion layer. For the yearlong analysis, BLHs were determined using
the Heffter algorithm (Heffter, 1980), as these BLHs agreed with sonde measurements during the study, are independent of
cloudiness, and have been used for analysis at this site previously (Ghate et al., 2023). Site-measured horizontal cloud fractions
($CF_H$) were determined from the percentage of opaque pixels in total sky imager hemispheric sky images recorded every
minute during daylight hours and when solar elevation was above 10° (Flynn and Morris, 2022). Site-measured vertical cloud
fractions ($CF_V$) and 3-dimensional cloud fractions ($CF_{3A}$) were calculated according to Eq. 2 and 3, respectively. For this
calculation, sonde-derived BLHs were linearly interpolated to match the time points of CBH measurements, resulting in $CF_V$
uncertainty largely dependent on assignment of BLH. This calculation assumed the vertical distance between the detected
CBH and BLH was fully filled with cloud, such that the cloud horizontal depth in $CF_H$ does not impact the calculation.

$$CF_V = \frac{BLH-CBH}{BLH} \quad\quad\quad (2)$$

$$CF_{3A} = CF_H \times CF_V \quad\quad\quad (3)$$

$CF_{3A}$ was compared to 3-dimensional cloud fraction derived from MERRA-2 (Modern-Era Retrospective Reanalysis for
Research and Applications, Version 2) (Gelaro et al., 2017) ($CF_{3M}$), which resolves cloud properties and cloud fraction
($CF_{VerticalLayer}$) at 0.5° × 0.625° resolution for 42 vertical pressure-resolved layers ($\Delta P_{VerticalLayer}$) every three hours. $CF_{3M}$ was



calculated as a weighted average cloud fraction (Eq. 4) over the entire boundary layer ($\Delta P_{BoundarylLayer}$) for a 4° × 5.625° region encompassing Graciosa Island (coordinates 37-41 °N and 30.625-25 °W). Boundary layers used for calculation were again based on the Heffter algorithm for sonde measurements (Riihimaki et al., 2022) and linearly interpolated to match MERRA-2 time points. These latitude, longitude, and boundary layer constraints were chosen to align with the inputs into the global chemical transport model, GEOS-Chem, if one were to model the impact of cloud chemistry in this region (Holmes et al., 2019).

$$CF_{3M} = \frac{CF_{VerticalLayer} \times \Delta P_{VerticalLayer}}{\Delta P_{Boundary\ Layer}} \tag{4}$$

## 3 Results

### 3.1 Cloud and gas-phase sulfur measurements at ENA

The time series of DMS, MeSH, HPMTF, and $CF_{3A}$ are shown in Fig. 1. DMS showed large variability throughout the study, mostly driven by wind speed, and averaged $106 \pm 69$ ppt. Here and in the following reported measurements of gas-phase concentrations and gas-phase ratios, standard deviations reflect natural variability in ambient concentrations. MeSH closely tracked DMS throughout ($R^2 = 0.56$), indicative of their shared DMSP source. However, MeSH concentrations were roughly a factor of five lower, averaging $16 \pm 13$ ppt. Both DMS and MeSH were highest in the early mornings hours when their oxidative loss was at a minimum and lowest in the afternoons. The average and interquartile range of the nighttime concentration ratio (Hr. 0-7) of [DMS]/[MeSH] was 6.5 (4.3-7.5) (Fig. 2). This is in line with existing measurements of the emission flux ratio of $E_{DMS}/E_{MeSH}$ in coastal Southern California ($5.5 \pm 3.0$) (Novak et al., 2022) and in the Southwest Pacific (3-7) (Lawson et al., 2020), and concentration ratio measurements of [DMS]/[MeSH] in a low oxidant mesocosm experiment during typical coastal ocean biological conditions ($4.60 \pm 0.93$) (Kilgour et al., 2022).

HPMTF measured significantly lower in concentration than its precursor DMS, with a 24-hour average of $0.7 \pm 1.1$ ppt and afternoon average (Hr. 13-17) of $1.6 \pm 1.7$ ppt. The median and interquartile ranges of [DMS]/[HPMTF] across all data points were 317 (73-1797). HPMTF also exhibited a strong diurnal profile, peaking in the late afternoons between the hours of 13 and 17 and was mostly at or below the detection limit in the nights and early mornings. Its near-zero concentrations in the early mornings suggested that HPMTF production from DMS restarted daily. The afternoon maximum in HPMTF and minimum in DMS resulted in low and stable [DMS]/[HPMTF] ratios during the afternoons, which are exploited in the box model analysis in Section 3.2.2.



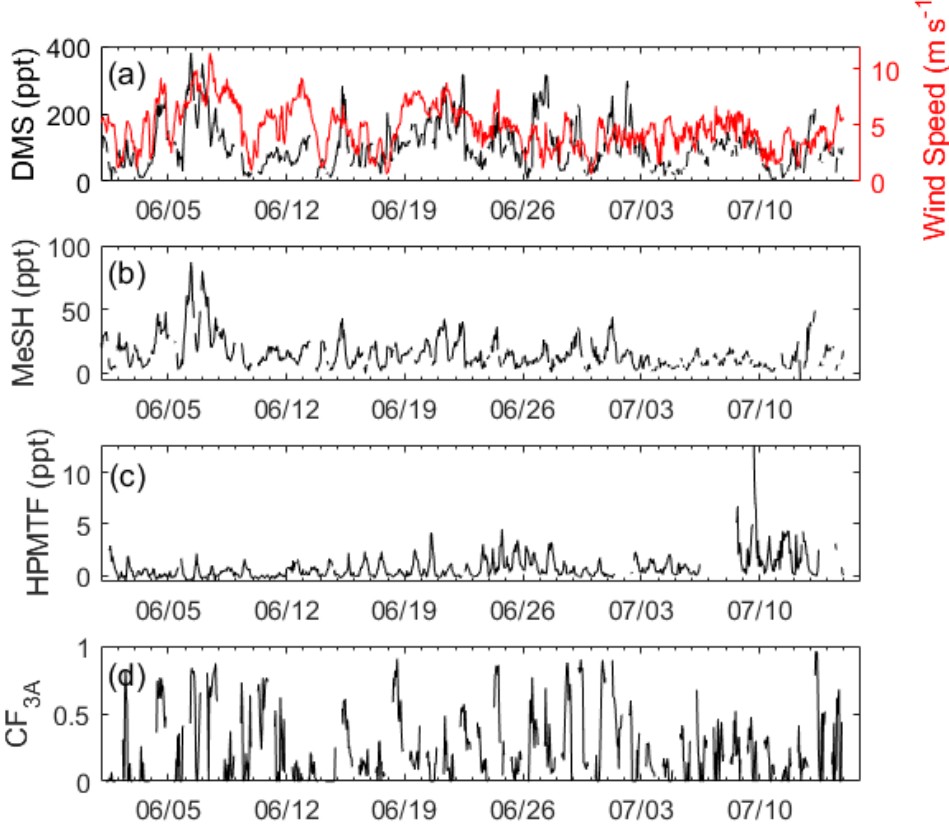

**Figure 1: Hourly averaged time series of (a) DMS measured by RT-Vocus and wind speed, (b) MeSH measured by RT-Vocus, (c) HPMTF measured by Vocus AIM with iodide reagent ions, and (d) site-measured 3D cloud fraction ($CF_{3A}$) calculated according to Eq. 3.**

$CF_{3A}$ during the study averaged $0.28 \pm 0.27$, where the standard deviation reflects natural variability in cloud cover. Numerous

time points of $CF_{3A}$ were near 1, indicating full cloud filling the region horizontally and vertically within the boundary layer.

Maximum daily $CF_{3A}$ occurred in the morning and steadily declined throughout the day into the evening. The sky imager used

to measure $CF_H$ only collected data during daylight, resulting in no nighttime information on $CF_{3A}$ at this site. The sky imager

cloud mask retrieval was also optimized for later in the day, which could lead to false or exaggerated clouds in the dusk and

dawn and might have influenced the peak $CF_{3A}$ in the mornings. During the entire study, [DMS]/[HPMTF] exhibited a weak

positive correlation with $CF_{3A}$ ($R^2 = 0.27$) (Fig. S3).





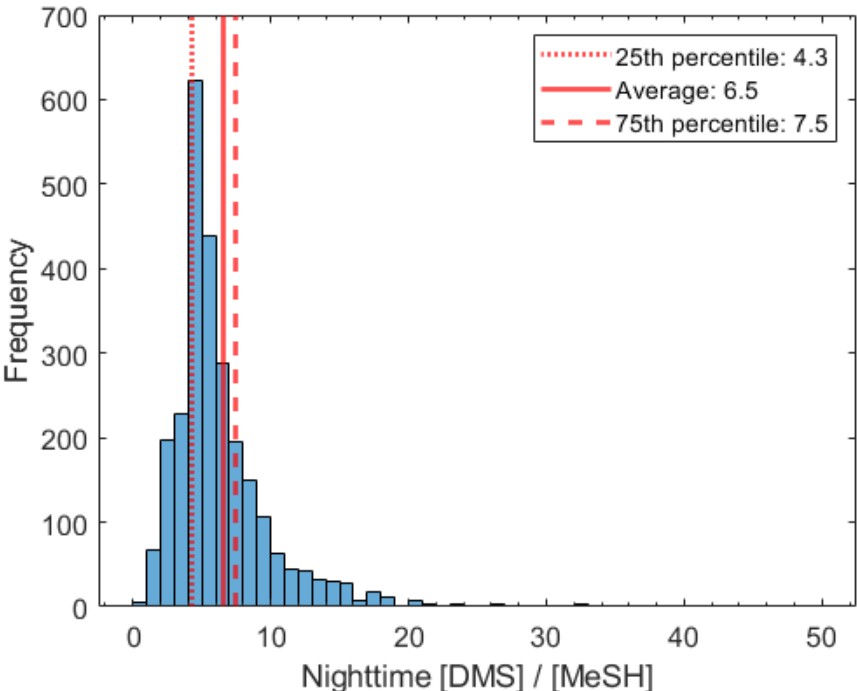

**Figure 2: Histogram of five-minute averaged [DMS]/[MeSH] ratio during Hr. 0-7 when oxidative loss was at a minimum.**

### 3.2 Measurement-constrained box model to assess cloud loss rates

### 3.2.1 Sensitivity to meteorological and chemical constraints

The model was run with a constant temperature diurnal profile corresponding to the diurnally-averaged measurements during the study, averaging 292 K. At this temperature, 61% of DMS OH-oxidation occurred by H-abstraction, which could later form HPMTF, and 39% occurred by OH-addition with no potential formation of HPMTF (Fig. S4). The diurnal-average temperature-dependent MTMP isomerization rate forming HPMTF was 0.036 s$^{-1}$ (Assaf et al., 2023). The ambient temperature in this study was lower than for which this rate constant was experimentally measured (314-433 K) and is calculated based on the extrapolation in Assaf et al. (2023). Model NO, HO$_2$, and RO$_2$ concentrations at Hour 15 were 2, 12, and 36 ppt, respectively, resulting in an $\alpha_{HPMTF}$, defined previously as the yield of HPMTF from the DMS H-abstraction pathway, of 0.85. Running the model with a time-varying temperature corresponding to the observed range over six weeks in the study (minimum 288 K - maximum 296 K) would result in a 31% increase in the diurnally-averaged HPMTF production rate and 9% decrease in afternoon [DMS]/[HPMTF].



Additionally, at a boundary layer height of 1000 m and exchange velocity of 0.5 cm s$^{-1}$ between the boundary layer and free troposphere, 74% of DMS in the model was oxidized in the boundary layer and 26% was oxidized in the free troposphere above. The fraction of DMS oxidized in the boundary layer is highly dependent on the boundary layer height and exchange velocity (Fig. S5), both of which have considerable uncertainty in the marine atmosphere, where boundary layers can be stable, without a strong inversion layer, and mixing between the free troposphere is difficult to measure (Faloona, 2009). The following analysis and discussion represent DMS oxidation in the Azores-region MBL. The lower temperature in the free troposphere would shift DMS OH-oxidation toward OH-addition and slow down the MTMP isomerization rate forming HPMTF (Assaf et al., 2023). At a representative summertime free troposphere temperature in this region of 283 K, the percentage of DMS OH-oxidation occurring by H-abstraction would reduce to 46% and the MTMP isomerization rate would slow to 0.016 s$^{-1}$; both reductions indicate the production of DMS-derived SO$_2$ and OCS would be lower in the free troposphere than in the MBL.

### 3.2.2 Evaluation of HPMTF loss rates

Figure 3a shows modelled [DMS]/[HPMTF] for several cloud loss rates. Cloud loss was modelled as a constant first order sink with respect to HPMTF concentration. In the modelled clear sky, where HPMTF was only lost by gas-phase oxidation, aerosol uptake, and deposition, [DMS]/[HPMTF] ranged between 4.2 and 12.7 during the course of a day. [DMS]/[HPMTF] in the afternoon, between hours 13 and 17, averaged 4.5 ± 0.5 in the modelled clear sky. This afternoon range corresponded to the diurnal maximum in HPMTF concentration and diurnal minimum in [DMS]/[HPMTF]. Since HPMTF was often at or below the detection limit in the nighttime, Hr. 13-17 are used to interpret model-measurement comparison. Furthermore, the near-zero nighttime HPMTF concentrations throughout the study meant HPMTF production restarted daily with OH production, suggesting only cloud cover along the air mass back trajectory in the hours between sunrise and the end of the model-measurement comparison period (Hr. 17) impacted HPMTF chemistry in the model. Since no consistently clear sky day existed during the study, a close case, occurring on July 11, was used to assess how well the model captured HPMTF chemistry in low cloud fraction conditions. Average CF$_{3A}$ on July 11 between Hr. 6 and 17 was 0.065 ± 0.055 (Fig. 3b, Fig. 3c), and the average afternoon [DMS]/[HPMTF] was 7.6 ± 1.1. This is slightly above the clear sky [DMS]/[HPMTF] (Fig. 3a), which could be due to a small amount of cloud cover overhead, cloud presence along the trajectory prior to the afternoon [DMS]/[HPMTF] comparison period, and/or uncertainty in the non-cloud HPMTF loss rates. Nonetheless, the close agreement indicates that DMS-HPMTF chemistry in the model is reasonably well-captured for the clear sky case and highlights that a large HPMTF loss to photolysis is not necessary, as has been implied previously (Khan et al., 2021).

The HPMTF cloud loss term was determined for all study days. The rate of cloud loss was determined as the value needed to make up the residual difference between the modelled clear sky [DMS]/[HPMTF] in the afternoon and the measurements of afternoon [DMS]/[HPMTF]. For example, to match the observed [DMS]/[HPMTF] on July 11, a small cloud loss term of 0.24 h$^{-1}$ was required. This is also shown for two additional days in Fig. 3, July 3 and June 27. For July 3, a cloud loss term of 0.94



h$^{-1}$ was needed for the model to match the measured afternoon [DMS]/[HPMTF] of 20 ± 3. For June 27, a cloud loss term of 2.5 h$^{-1}$ was needed for the model to match the measured afternoon [DMS]/[HPMTF] of 49 ± 12. As expected, days with the faster cloud loss rates have higher $CF_{3A}$, where the $CF_{3A}$ between Hr. 6 and Hr. 17 was 0.17 ± 0.10 for July 3 and 0.29 ± 0.22 for June 27.


The days shown in Fig. 3 were selected as case studies as they displayed a range in $CF_{3A}$ and had unstable, well-mixed boundary layers based on vertical profiles in potential temperature and water vapor mixing ratio (Fig. S6). Unstable, well-mixed boundary layers occurred on 16 of the 31 study days. During these conditions, we expect that the ground-based measurements of [DMS]/[HPMTF] are similar to [DMS]/[HPMTF] throughout the boundary layer due to strong vertical mixing. As a result,

the inferred cloud loss terms from [DMS]/[HPMTF] for these days is interpreted as an estimate of the cloud entrainment rate, thought of as the mixing rate of clear air into cloud. The three days in Fig. 3 were also chosen because they had more uniform cloud fraction during the time of HPMTF production. In an ideal case, where cloud fraction and cloud type are constant, measured [DMS]/[HPMTF] would fall exactly along the modelled [DMS]/[HPMTF]. In practice, $CF_{3A}$ varies throughout the day (Fig. 3b) due to changes in boundary layer height and horizontal cloud cover, and the GOES images (NOAA National

Centers for Environmental Information, 2017) in Fig. 3cde represent only one snapshot in time. It is more likely that the HPMTF cloud loss term changes throughout the day as $CF_{3A}$ in the sampling region evolves. As a result, extending this method to all study days incorporates some uncertainty in the derived cloud loss terms due to the cloud field at the site changing during the time period of HPMTF production.

Following the approach outlined above, we derive cloud loss rates based on the residual loss required for modelled [DMS]/[HPMTF] to equal the measured [DMS]/[HPMTF] ratio for all 31 days. This approach enables cloud loss terms to be derived over long time periods and significantly increases the data coverage compared to prior work. However, the indirect methodology of assigning cloud loss terms based on residual differences in [DMS]/[HPMTF] from a base case means any inaccuracy in the model (e.g. rates of other HPMTF loss processes, variable fractions of DMS forming HPMTF based on

changing temperature and NO and $RO_2$ concentrations) can contribute to uncertainty in the derived cloud loss terms. Additionally, we are utilizing near-surface measurements of [DMS] and [HPMTF], which is less ideal compared to making vertically-resolved measurements of these species, or direct airborne eddy covariance flux measurements at different altitudes to directly calculate HPMTF loss terms (Novak et al., 2021). Figure 4 displays the loss rates of HPMTF to cloud, aerosol, dry deposition, and OH for each day following this approach. Aerosol uptake and dry deposition accounted for a minor fraction of

HPMTF loss during this study, while gas-phase oxidation by OH and loss to cloud made more significant contributions. Bracketing the modelled outputs based on the uncertainty in cloud loss rates derived from [HPMTF] uncertainty, on average, 79-91% of HPMTF in the model was lost to cloud, 7-16% was lost to OH, and the remaining 2-6% was lost to aerosol and dry deposition. The reported loss of HPMTF to aerosol is likely a lower limit as any acidity in the ambient marine aerosol (Angle et al., 2021) could cause enhanced HPMTF uptake (Liggio and Li, 2006). Additionally, we do not have concurrent





measurements of coarse mode (> 1 μm diameter) sea spray aerosol particles which are hygroscopic (Zieger et al., 2017) and could provide an enhanced surface area for HPMTF uptake, particularly during strong winds that promote sea spray production. However, this loss process is still expected to be small relative to cloud due to the large difference in surface area between aerosol and cloud droplets. HPMTF was lost to cloud chemistry at a rate approximately 5-13 times faster than to OH chemistry during this study. Regardless of the uncertainty in cloud loss rates, this analysis highlights that cloud loss is the dominant

HPMTF loss process in the MBL during this study. The median lifetime of HPMTF to cloud bracketed by uncertainties in HPMTF quantification were 0.29-0.81 h (interquartile range of 0.06-2.24 h), which were significantly faster than the instantaneous chemical lifetime to OH (greater than 4 h). This is consistent with airborne measurements in Novak et al. (2021) off the coast of Southern California, where the lifetime of HPMTF to cloud (1.2 ± 0.6 h) was also much faster than the lifetime to OH (greater than 5 h). Analysis in this work affirms cloud chemistry as the dominant HPMTF loss pathway now over a

longer time period and in another region of the MBL.

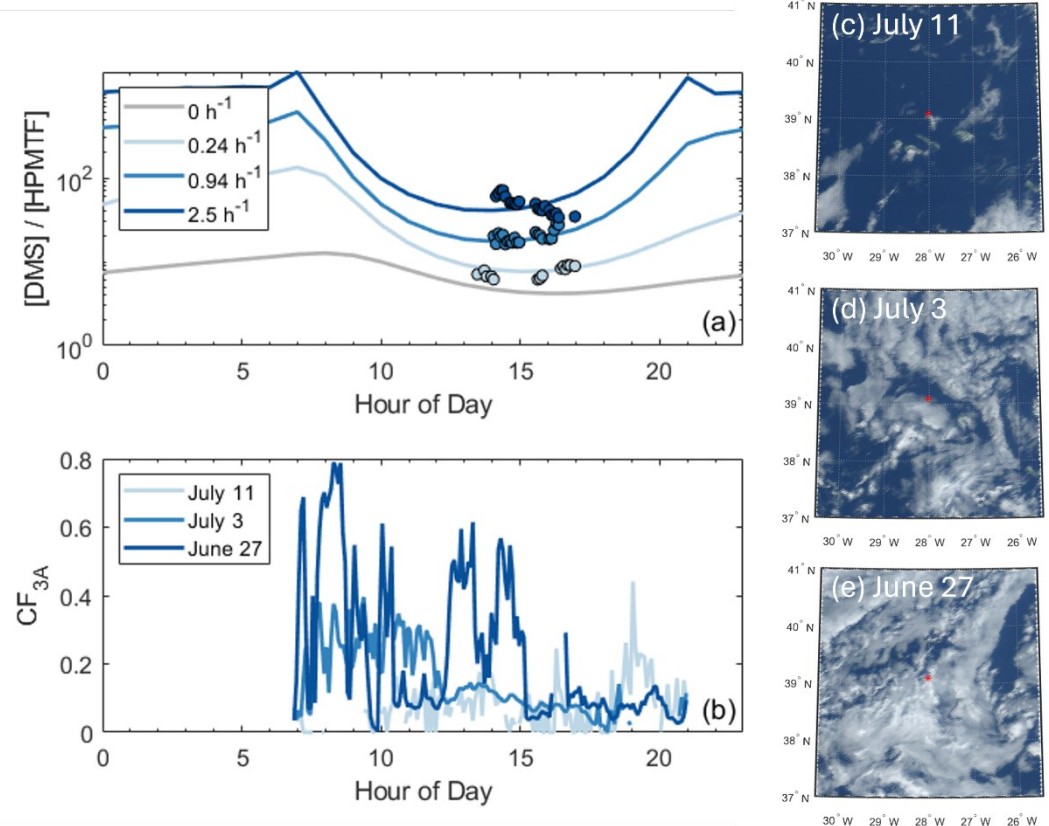

**Figure 3: (a) Modelled [DMS]/[HPMTF] for different HPMTF cloud loss rates, using the formic acid calibration factor for HPMTF, which represents upper limits on cloud loss rates. The gray line with a cloud loss term of 0 h⁻¹ represents the modelled clear sky. Scattered points represent 5-minute averaged [DMS]/[HPMTF] measurements for colors**
**matching the dates in (b). (b) Site-measured 3-dimensional cloud fraction (CF₃A) for three selected days during the study. GOES imagery for 4° × 5° regions around Graciosa Island at Hour 15 for (c) July 11, (d) July 3, and (e) June 27. The approximate measurement location on Graciosa Island (39.0916° N, 28.0257° W) is marked with a red asterisk.**



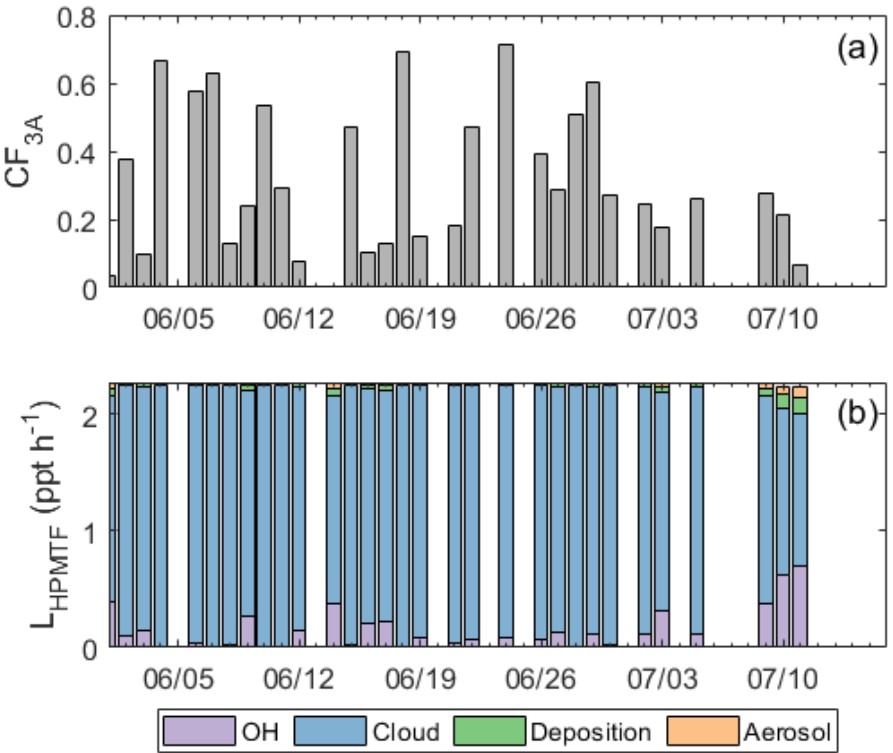

**Figure 4: (a) Site-measured 3-dimensional cloud fraction (CF$_{3A}$) for days with modelled HPMTF loss rates. (b) Histogram of modelled HPMTF loss rates for the 31 study days, separated by HPMTF loss to OH, cloud, deposition, and aerosol. Breakdown of HPMTF loss rates corresponds to the model run with a formic acid calibration factor for HPMTF, resulting in an upper limit on the fraction of HPMTF lost to cloud.**

## 4 Discussion

### 4.1 Dependence of cloud loss rates on site-measured 3-dimensional cloud fraction

The HPMTF cloud loss rates inferred from the model-measurement comparison of [DMS]/[HPMTF] for all 31 study days are scattered against CF$_{3A}$ in Fig. 5 and colored by afternoon relative humidity. Higher cloud loss rates of HPMTF were observed on days with increased relative humidity, indicative of sampling in a cloud-filled boundary layer (Chernykh and Eskridge, 1996). Additionally, since the relative humidity across all days, even at low CF$_{3A}$, was above the efflorescence point of inorganic sea spray aerosol (50%) (Zieger et al., 2017), we expect the variability in inlet loss (assuming the inlet is coated in wet sea spray aerosol) was minimal and take the observed trend to be robust. Across all days, the median cloud loss rate of HPMTF to cloud, bracketed by uncertainties in HPMTF quantifications, was 1.2-3.4 h$^{-1}$ (interquartile range 0.45-19 h$^{-1}$). In just the days with unstable, well-mixed boundary layers demarcated with squares in Fig. 5, the median and interquartile ranges of cloud loss rates were 0.86-2.5 h$^{-1}$ and 0.34-9.2 h$^{-1}$, respectively. Inferring HPMTF cloud loss rates from ground-based



measurements of [DMS]/[HPMTF] relies on the assumption that the near-surface [DMS]/[HPMTF] measurements are
representative of [DMS]/[HPMTF] throughout the boundary layer. While this is a fair assumption in well-mixed boundary
layers with strong vertical mixing, this is likely not the case in stable boundary layers. In well-mixed boundary layers, the rate
of HPMTF cloud loss can be thought of as the entrainment of HPMTF in clear air into cloud, which has previously been
estimated at 1 $h^{-1}$ for stratocumulus clouds based on large eddy simulation studies (Feingold et al., 1998). Our derived cloud
loss rates on the well-mixed days are closer to these values. The dashed black line in Fig. 5 shows the predicted cloud loss of
HPMTF based on an average 1 $h^{-1}$ entrainment rate. While our HPMTF cloud loss rates are faster than those predicted, they
follow the same shape where cloud loss increased and saturated with increasing $CF_{3A}$. Unlike the prediction, the derived
HPMTF cloud loss rates were variable for individual $CF_{3A}$. This is likely a result of heterogeneity in cloud fraction (Fig. 3b)
and cloud type during the time period of HPMTF production, which are ignored in the calculation of the predicted rate. One
particularly important aspect of cloud heterogeneity affecting the calculations can come in instances of near-complete vertical
cloud fraction, as HPMTF was likely lost to cloud at the diffusion limit, which is not well-captured by the inferred cloud loss
from $CF_{3A}$ alone and can result in fast cloud loss rates.



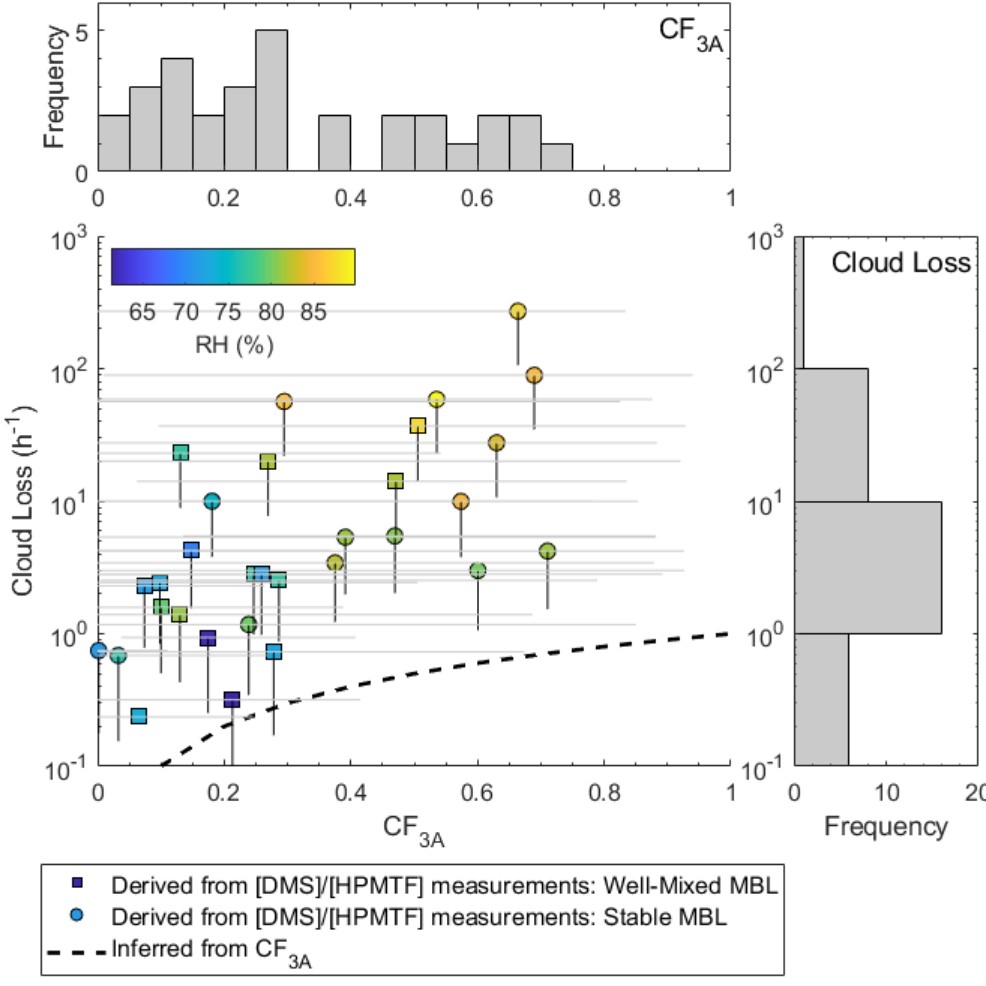

**Figure 5: Relationship between derived cloud loss rates and $CF_{3A}$ measured at the site. Scattered data points represent derived cloud loss terms based on the residual in [DMS]/[HPMTF], where HPMTF was calibrated with the formic acid**
**calibration factor, and are colored by afternoon relative humidity in Hr. 13-17. Square points represent days with a well-mixed MBL, defined by near-vertical slopes in potential temperature and water mixing ratio, and circles represent a stable, not well-mixed MBL, defined by non-zero slopes in potential temperature and water mixing ratio. Black vertical lines below the points represent the range in cloud loss rates based on HPMTF uncertainty. Gray horizontal lines across the points represent the minimum and maximum $CF_{3A}$ during the time of HPMTF production from Hr. 6-**
**17. The black dotted line represents an expected cloud loss term calculated as the product of $CF_{3A}$ and an assumed 1 h$^{-1}$ entrainment rate. The variability in $CF_{3A}$ and cloud loss are also shown as histograms on the mirrored axes.**

## 4.2 Impacts of cloud loss on MBL DMS conversion to SO₂ and OCS

The efficient removal of the HPMTF intermediate via cloud chemistry has a correspondingly large impact on the amount of

$SO_2$ and OCS derived from DMS oxidized in the MBL. Figure 6 shows the fraction of DMS converted to $SO_2$ and OCS in the



modelled clear sky case (black dashed line) and when implementing the derived cloud loss rates from [DMS]/[HPMTF] (blue histogram). In both cases, the converted fraction is calculated as the 24-hour average of the production rate of the product from DMS divided by the chemical loss rate of DMS. In clear sky conditions, 27% of DMS oxidized in the MBL is converted to $SO_2$ (Fig. 6a) and 3% of DMS oxidized in the MBL is converted to OCS (Fig. 6b). This analysis indicates that HPMTF cloud loss, at rates derived from [DMS]/[HPMTF], could decrease MBL DMS-derived $SO_2$ by on average 52-60% (Fig. 6a) and

MBL DMS-derived OCS by on average 80-92% (Fig. 6b), where the ranges correspond to uncertainty in cloud loss terms derived from HPMTF quantification uncertainty. These findings are consistent with earlier, global modeling work based on an HPMTF cloud loss rate determined from a flight off the coast of Southern California (Novak et al., 2021) that showed HPMTF cloud loss reduced global DMS-derived $SO_2$ production by 35% (Novak et al., 2021) and OCS production by 92% (Jernigan et al., 2022a).


The oxidation of $SO_2$ to sulfuric acid has been shown to result in new particle formation in the MBL (Covert et al., 1992). Reduction in MBL $SO_2$ due to cloud chemistry shown here suggests that nucleation and growth rates of new particles in the MBL might be slower than previously thought, or non-$SO_2$ precursors, such as ammonia (Jokinen et al., 2018) and iodine-containing molecules (Baccarini et al., 2020) might play larger roles than once believed, especially in cloudy regions.

Importantly, aqueous phase HPMTF chemistry in cloud has been shown to promptly form $SO_4^{2-}$ at unit yield (Jernigan et al., 2024). Including prompt sulfate production from HPMTF cloud chemistry in our model leads to the production of 0.18-0.20 $\mu$g m$^{-3}$ $SO_4^{2-}$ daily for the median derived cloud loss rate of 1.2-3.4 h$^{-1}$. This means that although HPMTF cloud chemistry largely reduces $SO_2$ concentrations in the MBL, sulfate aerosol is still being formed in cloudy regions, albeit through a different mechanism. Additionally, the DMS oxidation products along the OH-addition pathway (DMSO, $DMSO_2$, MSIA, and MSA)

are also soluble. If they are irreversibly lost to cloud like HPMTF, then cloud chemistry could even more drastically control the production of DMS-derived products. Lastly, by decreasing the amount of OCS produced from DMS in the MBL, HPMTF cloud loss reduces the amount of OCS that is transported to the stratosphere (Montzka et al., 2007), where it can serve as a precursor to stratospheric $SO_4^{2-}$ and control Earth's radiative budget (Brühl et al., 2012; Kremser et al., 2016).

**4.3 Comparison between DMS oxidation using derived cloud loss rates and current implementations of DMS oxidation**
**in global models**

The conversion of MBL DMS to $SO_2$ and OCS incorporating cloud loss rates derived from *in situ* measurements of [DMS]/[HPMTF] are compared to the conversions of DMS to $SO_2$ and OCS from DMS oxidation in common global model implementations. Without incorporating HPMTF chemistry, the global chemical transport model, GEOS-Chem, assigns direct, fixed yields of $SO_2$ (1, 1, 0.75) and MSA (0, 0, 0.25) from $NO_3$-oxidation, OH-oxidation H-abstraction, and OH-oxidation

OH-addition pathways, respectively (Chin et al., 1996). At the limit of no cloud present, this yield implementation results in 93% of MBL DMS converted to $SO_2$ in the F0AM box model, which is significantly larger than the amount of $SO_2$ formed when implementing the [DMS]/[HPMTF] derived cloud loss rates. The historical model of OCS formation from DMS OH-



oxidation also uses a direct, fixed yield of 0.007 (Barnes et al., 1994), which aligns with our findings here in the presence of cloud.


As introduced earlier, when taking into account cloud chemistry, GEOS-Chem parameterizes cloud loss of reactive, soluble gases as the product of the boundary layer grid cell MERRA-2 cloud fraction ($CF_{3M}$) (Global Modeling and Assimilation Office (GMAO), 2015) (Holmes et al., 2019) and an average entrainment rate of 1 h$^{-1}$ based on large eddy simulation studies of stratocumulus clouds (Feingold et al., 1998). This cloud loss term that would be input into GEOS-Chem was run in the

developed F0AM box model, with HPMTF chemistry, to assess how closely it matched the modelled outputs from the derived cloud loss terms; the results of this analysis are shown by orange bars in Fig. 6. Utilizing a cloud loss rate for HPMTF determined by $CF_{3M}$ resulted in on average, a factor of 1.7-2.1 and 3.7-9.5 more MBL DMS-derived $SO_2$ and OCS, respectively, compared to the derived [DMS]/[HPMTF] cloud loss rate implementation, where the ranges indicate propagated uncertainty from HPMTF concentrations. These MERRA-2-based values were closer to the fractions of $SO_2$ and OCS formed

from DMS in the modelled clear sky case. This is partly due to the consistent underestimate by up to a factor of four of $CF_{3M}$ relative to $CF_{3A}$ during June and July (Fig. 7). This is consistent with significant errors in the accuracy of satellite-derived cloud fractions in the MBL where the boundary layer is low and where there is persistent cloud cover (Kuma et al., 2020). Our analysis using *in situ* derived cloud loss rates and site measurements of 3D cloud fraction suggest that (1) cloud processing in models is required to accurately capture the fate of DMS, and (2) cloud loss parameterized by satellite-retrievals of low-level



cloud fraction underestimate the effects of HPMTF cloud chemistry. Models might better capture the impacts of HPMTF cloud chemistry by assuming full conversion to $SO_4^{2-}$ when cloud is present and conversion to $SO_2$ in clear sky conditions.

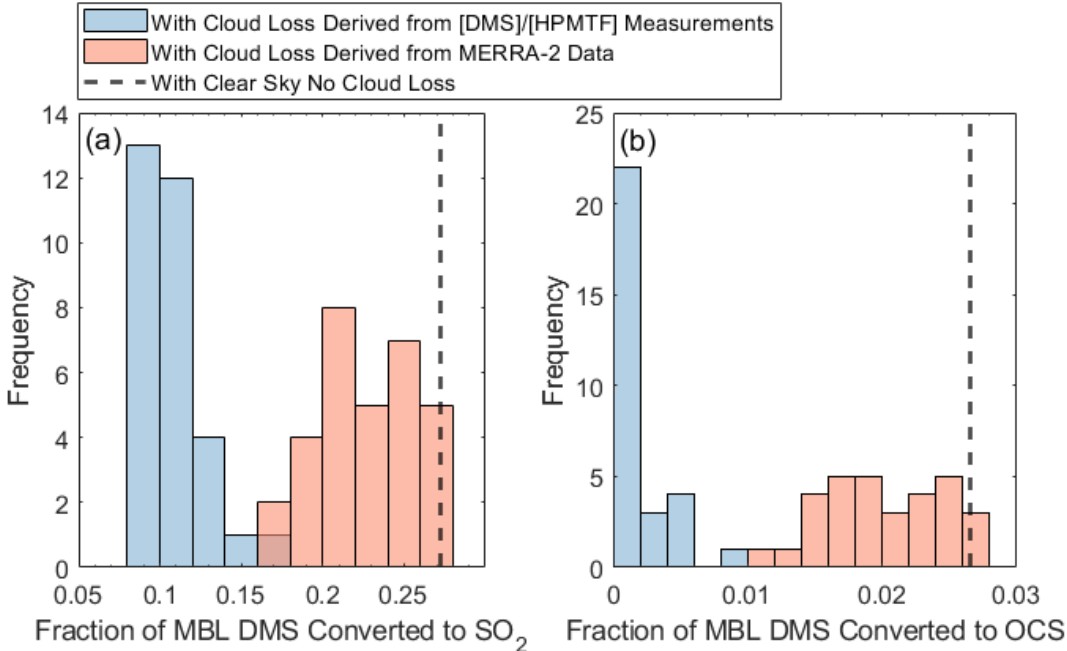

**Figure 6: Fraction of MBL DMS converted to (a) SO₂ and (b) OCS shown for the derived cloud loss rates based on residual [DMS]/[HPMTF] and based on the MERRA-2 cloud fractions and a 1 h⁻¹ entrainment rate. The modelled clear**
**sky conversion, using HPMTF chemistry and heterogeneous loss of DMS oxidation products, is in black.**

**4.4 Insights into DMS-oxidation year-round**

*In situ* DMS, HPMTF, and $CF_{3A}$ measurements show fast cloud processing strongly regulates the fate of HPMTF, to a greater extent than what's currently prescribed in global models, during the study in June and July of 2022. Using DMS climatology and year-round measurements made at ENA, we build upon the summertime chemical box model to speculate on the role of
cloud processing in DMS-oxidation in this region during time periods beyond the summer intensive period. Monthly-averaged DMS fluxes were taken from Hulswar et al. (2022) climatology for an 8° radius box around Graciosa, approximating the DMS lifetime. Monthly OH profiles were determined in F0AM using the hybrid method for calculating photolysis frequencies with default surface albedo and $O_3$ column (Wolfe et al., 2016), and validated by the OH climatology in Spivakovsky et al. (2000). Monthly-averaged site measurements of meteorological (pressure, temperature, and relative humidity) (Uin et al., 2019) and
boundary layer height data (Heffter, 1980) were used as inputs. Remaining trace gas constraints, dilution terms, and non-cloud HPMTF loss processes were kept constant from the summertime model.

In the model, we demonstrate that the clear sky HPMTF concentration in fall and winter months is reduced relative to its concentration in spring and summer months, in line with our understanding of its production as a function of DMS and oxidant



concentration, temperature, and boundary layer height. Dissolved DMS concentrations are highest in this region during the spring and summer months (<6 nM), and are low in other months (<2 nM) (Hulswar et al., 2022). Similarly, air temperature in the MBL is lower in months outside of this study period, with sonde profiles at ENA indicating a wintertime MBL temperature of 10 °C is representative. At this temperature, only 46% of DMS OH-oxidation occurs by H-abstraction (compared to 61% in the summertime model), and the MTMP isomerization rate is approximately halved. Lower daylight

hours in non-summer months reduce OH concentrations; global OH climatology indicates surface OH at this latitude is close to a factor of 10 lower in January compared to July (Spivakovsky et al., 2000). Furthermore, modelled NO and $RO_2$ concentrations are roughly a factor of two higher in July than in January, though still low, resulting in $\alpha_{HPMTF}$ of 0.86 in July and $\alpha_{HPMTF}$ of 0.89 at Hour 15 in January. BLHs are less sensitive to seasonality at this site, with average wintertime BLHs roughly 20% higher than typical heights observed during June and July. The low precursor DMS concentration, increased

preference for the DMS OH-addition channel, slower MTMP isomerization rate, and reduced oxidative conditions result in modelled HPMTF production being approximately six times slower in January compared to July.

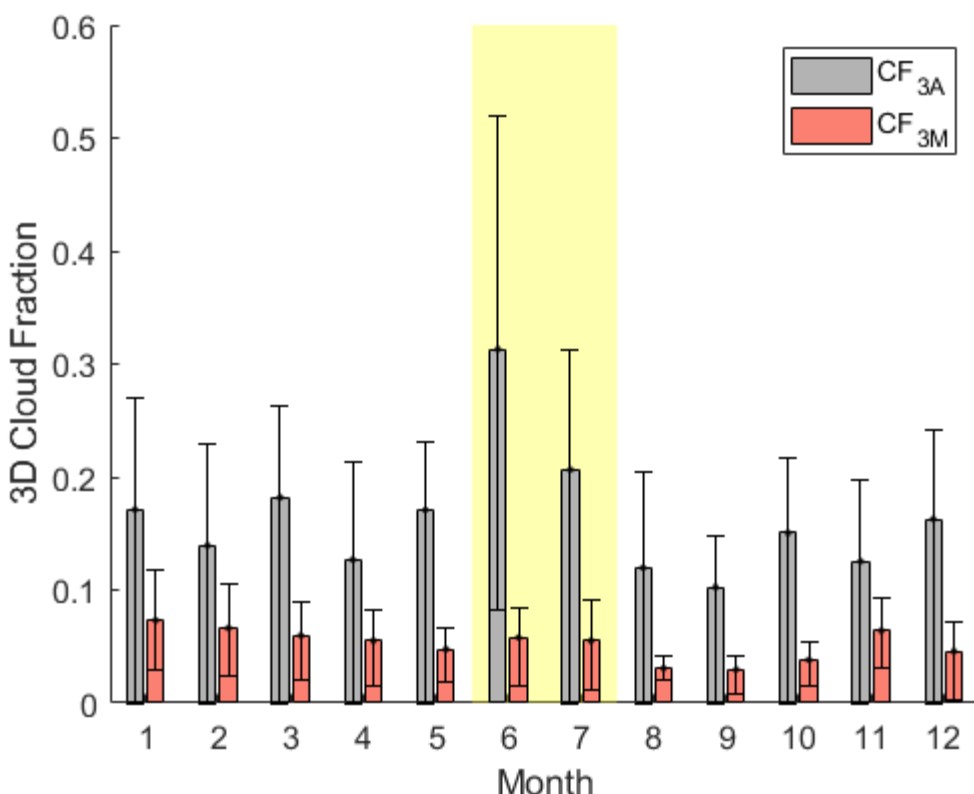

**Figure 7: Monthly mean boundary layer 3D cloud fraction measured at the site by ARM in gray (CF$_{3A}$) and measured by MERRA-2 (CF$_{3M}$) surrounding Graciosa in red. Error bars represent the interquartile ranges. Both CF$_{3A}$ and CF$_{3M}$**
**are calculated based on the boundary layer height determined by the Heffter approximation. The yellow shaded region indicates the time period of this study.**





While HPMTF production is lower beyond June and July of this study, measurements of $CF_{3A}$ demonstrate 3D cloud fraction at ENA is large year-round, shown in Fig. 7. $CF_{3A}$ peaks in June during this study, but other months in the year are consistent

with $CF_{3A}$ observed in July, during which some of our DMS and HPMTF measurements were made. Applying the median cloud loss rates derived from summertime [DMS]/[HPMTF] (1.2-3.4 $h^{-1}$) to all months, we demonstrate a low fraction of DMS ultimately forms $SO_2$ (3-12%) and OCS (<0.4%) at this site year-round (Fig. 8). Furthermore, applying a monthly-specific cloud loss rate derived from average $CF_{3M}$ from MERRA-2 and a 1 $h^{-1}$ entrainment rate indicates MERRA-2 continues to overestimate the amount of $SO_2$ and OCS produced from DMS across the entire year (Fig. 8). Together, these findings indicate

that cloud processing plays a large role in DMS-oxidation in the ENA MBL year-round, and persistent underestimates in 3D cloud fraction by MERRA-2 likely result in current global models underrepresenting the dominant impact of cloud chemistry across the entire year.

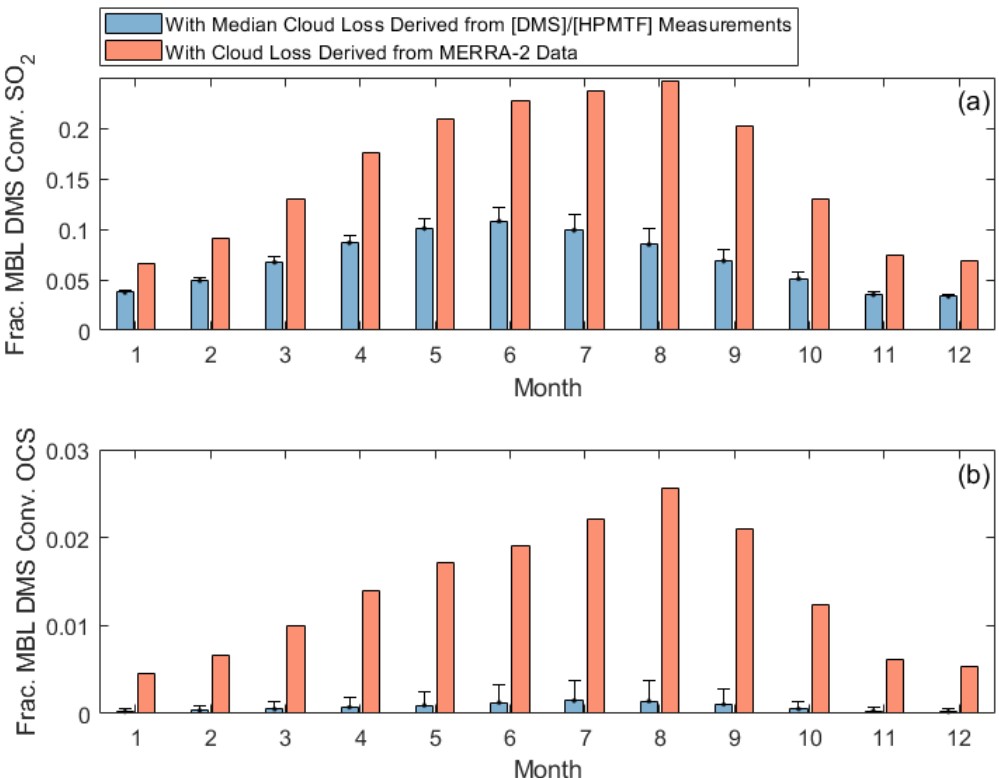

**Figure 8: Diurnally-averaged fractions of MBL DMS converted to (a) SO₂ and (b) OCS in the year-round model.**
**Derived cloud loss bars indicate running the model with the upper limit of the median cloud loss range (1.2-3.4 $h^{-1}$) and the error bar represents running the model with the lower limit of the range applied to all months. MERRA-2 cloud loss bars indicate running the model with monthly-specific cloud loss rates calculated as the product of the monthly averaged $CF_{3M}$ and a 1 $h^{-1}$ entrainment rate.**





Finally, we contextualize the impact of cloud processing on $SO_2$ production through a model test including MeSH. Assuming the loss rate of HPMTF to cloud derived in this study (1.2-3.4 $h^{-1}$ median) is representative of its loss rate in other seasons, then cloud chemistry can reduce the production of $SO_2$ from DMS by 49-67% year-round compared to the clear sky case, shown in Fig. 9. Given MeSH is an efficient MBL $SO_2$ source with a short lifetime to OH and its oxidation toward $SO_2$ does not proceed via the soluble HPMTF intermediate, it has the potential to further close the $SO_2$ budget. Utilizing a flux of MeSH

at 20% of the monthly-averaged DMS flux for this region (Hulswar et al., 2022), in line with the limited current measurements of flux ratios of DMS and MeSH (Lawson et al., 2020; Novak et al., 2022), shows MeSH (green in Fig. 9) can be competitive with DMS (purple in Fig. 9) as an $SO_2$ source in this region, where its oxidation has minimal temperature-dependence (Chen et al., 2023). MeSH is an especially important $SO_2$ source when accounting for cloud processing of HPMTF at the derived cloud loss rates, and in winter months, when preference for DMS OH-addition and slow MTMP isomerization limit $SO_2$

yielded from DMS-oxidation.

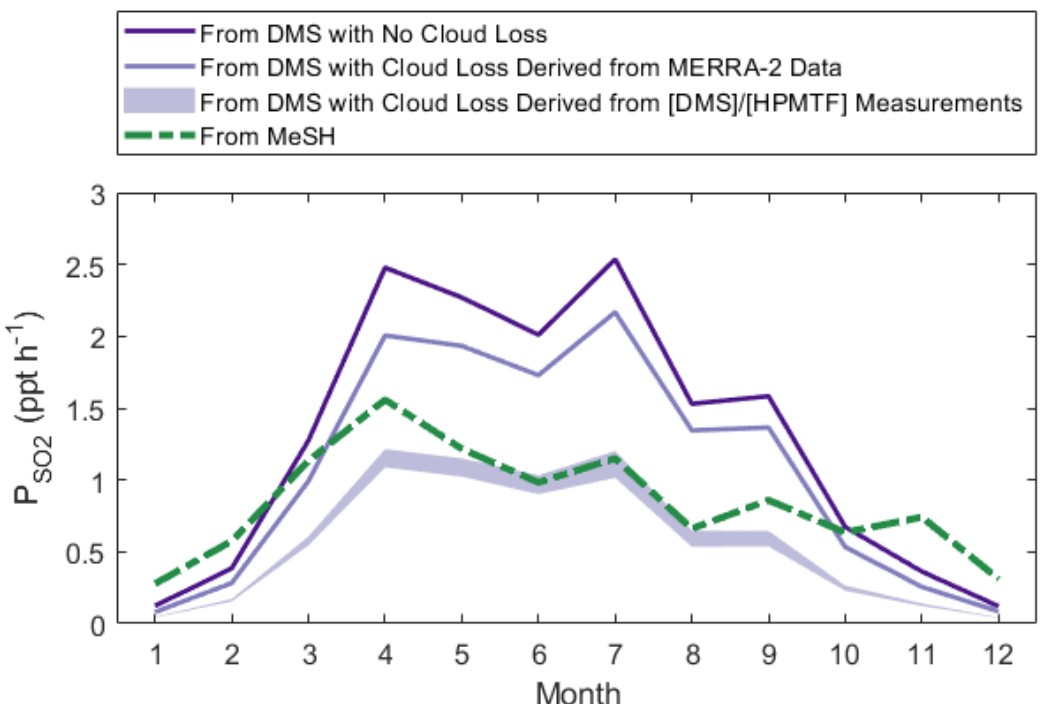

**Figure 9: Diurnally-averaged $SO_2$ production rates determined by the year-round model, where contributions to $SO_2$ from MeSH (green) and DMS (purple) are run individually. $SO_2$ production rates from DMS involving MERRA-2 cloud loss use monthly averaged $CF_{3M}$ and a 1 $h^{-1}$ entrainment rate, and production rates from DMS involving the**

**derived cloud loss rates from [DMS]/[HPMTF] use the median cloud loss rates (1.2-3.4 $h^{-1}$) applied to all months.**



## 5 Conclusions

This work utilizes measurements of the reactant and product pair, DMS and HPMTF, and a developed box model constrained by meteorological and trace gas measurements at the site to derive the loss rate of HPMTF to cloud in the ENA MBL during June and July 2022. This method was enabled by the considerable source of DMS from the oceans, and its large reservoir in the soluble oxidation product, HPMTF. The median derived cloud loss rate based on [DMS]/[HPMTF] analysis was 1.2-3.4 $h^{-1}$, leading to a median lifetime of HPMTF to cloud of 0.29-0.81 h. Box model analysis indicated cloud was the dominant sink of HPMTF, with on average, 79-91% of HPMTF lost to cloud, and 7-16% lost to the second strongest loss pathway, OH. Our findings are consistent with prior airborne flux analysis, where the HPMTF lifetime to cloud on a single flight leg was similarly fast ($1.2 \pm 0.6$ h) and similarly outpaced chemistry.

Our study demonstrates that cloud loss scaled with site-measured 3D cloud fraction over six weeks and controlled the fate of HPMTF in the MBL throughout this entire period. The chemically-derived cloud loss rates resulted in modelled reductions in DMS-derived MBL $SO_2$ and OCS of 52-60% and 80-92%, respectively. Since cloud processing sets MBL $SO_2$ and $SO_4^{2-}$ aerosol budgets from DMS, additional, highly sensitive measurements of MBL $SO_2$, DMS, and the other major marine $SO_2$ precursor, MeSH, are warranted to constrain drivers of $SO_2$ and new particle formation through the production of $H_2SO_4$. Lastly, this work utilizes DMS climatology and year-round measurements at ENA to suggest that cloud processing of HPMTF is important year-round in the ENA, beyond the measurement period, due to persistent boundary layer cloud cover. Since satellite products, like MERRA-2, retrieve low cloud fractions relative to *in situ* ground-based measurements, and global chemical transport models parameterize cloud loss by satellite cloud fraction, the controlling role of cloud processing in setting $SO_2$, OCS, and $SO_4^{2-}$ budgets is likely underrepresented in current global models.

### Data availability

DMS, MeSH, and HPMTF time series are available at http://digital.library.wisc.edu/1793/85493 (Kilgour and Bertram, 2024)

### Supplement

Contains details on the HPMTF measurement and uncertainty, supporting figures, and box model constraints.

### Author contributions

DBK, CMJ, and THB conceptualized the main ideas of the paper. DBK and CMJ collected RT-Vocus data, and processed and analyzed the data. SA and OG collected Vocus AIM data, with CM contributing to setup and supporting data analysis. OG analyzed Vocus AIM field data, and SA performed RH-dependent formic acid calibrations and analyzed calibration data. DBK



developed the box models and wrote the paper, with input from CMJ and THB. CM, MES, JW, JAT, PZ, and THB contributed

to AGENA campaign planning and execution, and supported data collection and analysis. All authors reviewed and edited the

paper.

**Competing interests**

At least one of the (co-)authors is a member of the editorial board of Atmospheric Chemistry and Physics.

**Financial support**

This material is based upon work supported by the U.S. Department of Energy, Office of Science, Office of Biological and

Environmental Research, Atmospheric System Research (ASR) under Award Number DE-SC0021985. This research used

resources of the Atmospheric Radiation Measurement (ARM) User Facility, which is a DOE Office of Science User Facility,

under Award Number AFC010011.

**Acknowledgments**

The authors thank the ARM staff at ENA and Los Alamos National Laboratory, including Bruno Cunha and Tercio Silva, for

their support and logistical contributions to the study, and the ARM instrument mentors, including Donna Flynn at Pacific

Northwest National Laboratory, for providing publicly accessible supporting data from ENA. Additional thanks to Ankur

Desai and Grant Petty at University of Wisconsin-Madison for valuable discussions on boundary layer stability. Lastly, the

authors acknowledge Glenn Wolfe for providing the F0AM box model, some of which was edited for use in this analysis, and

the tofTools team for providing tools for mass spectrometry data analysis.

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
