# Peer review of "Cloud processing of DMS oxidation products limits SO2 and OCS production in the Eastern North Atlantic marine boundary layer"

_EGUsphere, 2024_

## Referee Comment (RC2)

**Comment on manuscript egusphere-2024-1975**

August 28, 2024

**General comment**

Based on field measurements the authors present an assessment of how much the HPMTF loss to cloud affect the yields of SO2 and OCS from DMS oxidation. The manuscript is well written and conclusions sound. The analysis looks valid given the limited of knowledge about aqueous-phase oxidation of organo-sulfur compounds in the real atmosphere. I only have some remarks concerning how heterogeneous losses have been accounted for.

**Major comments**

- **Loss to aerosols**
  In Eq. 1 the expression for the loss frequency, $k$, to aerosols is given in a specific form which might not be familiar to every reader. It took me a while to find the proper reference [1] and assure myself it is correct. Please add that reference. The equation for $k$ is normally used with $\alpha$ instead of $\gamma$. For aerosol uptake the latter is more commonly used in an expression with only the second term as the cited DMS-relevant works [2, 3]. Why did the authors not used the usual form $1/4 \, v \, \gamma \, A$? A short explanation would be help the reader.

- **Loss to cloud droplets**
  It has been reported that the DMS loss due to aqueous-phase oxidation by O3 is not neglibible [4]. If considered by the authors, this would imply larger cloud loss of HPMTF in order to match the observed DMS/HPMTF ratios. Moreover, as it could be seen by the *gamma* values higher than for HPMTF, loss of DMSO, DMSO2, MSIA and MSA to clouds is likely affecting the SO2-yield as well. I understand the model used by the authors lacks explicit cloud chemistry but at least it would be good to have these points in the discussion of results.

**Minor comments**

- p. 3, l. 75: Upon O2 addition to the CH3S radical CH3SOO is produced and not CH3SO2. The latter comes later from CH3SOO isomerization and the T-dependent branching ratio between channels yielding either SO2 or MSA are for reactions involving CH3SO2. However, the CH3SOO isomerization is not included neither in MCM nor in the updated F0AM model as one can learn from Table S1. The author should consider how much of an impact it has on the overall SO2 conversion from DMS oxidation in their model.

- p. 18, l. 448: Saying that the results of this work "align" with the OCS-yield determined in the experiments by [5] is misleading. Those experiments were done in a chamber under non-atmospheric conditions and surely without liquid water. I would rather say it is a fortuitous coincidence that the two OCS-yields meant here are similar.

**References**

[1] D Jacob. Heterogeneous chemistry and tropospheric ozone. *Atmospheric Environment*, 34 (12-14):2131–2159, 2000. ISSN 13522310. doi: 10.1016/S1352-2310(99)00462-8. URL https://linkinghub.elsevier.com/retrieve/pii/S1352231099004628.

[2] Erik Hans Hoffmann, Bernd Heinold, Anne Kubin, Ina Tegen, and Hartmut Herrmann. The Importance of the Representation of DMS Oxidation in Global Chemistry-Climate Simulations. *Geophysical Research Letters*, 48(13):e2021GL094068, 2021. ISSN 1944-8007. doi: 10.1029/2021GL094068. URL https://agupubs.onlinelibrary.wiley.com/doi/abs/10.1029/2021GL094068. _eprint: https://agupubs.onlinelibrary.wiley.com/doi/pdf/10.1029/2021GL094068.

[3] Christopher M. Jernigan, Christopher D. Cappa, and Timothy H. Bertram. Reactive Uptake of Hydroperoxymethyl Thioformate to Sodium Chloride and Sodium Iodide Aerosol Particles. *The Journal of Physical Chemistry A*, June 2022. ISSN 1089-5639. doi: 10.1021/acs.jpca.2c03222. URL https://doi.org/10.1021/acs.jpca.2c03222. Publisher: American Chemical Society.

[4] Erik Hans Hoffmann, Andreas Tilgner, Roland Schrödner, Peter Bräuer, Ralf Wolke, and Hartmut Herrmann. An advanced modeling study on the impacts and atmospheric implications of multiphase dimethyl sulfide chemistry. *Proceedings of the National Academy of Sciences*, 113(42):11776–11781, October 2016. ISSN 0027-8424, 1091-6490. doi: 10.1073/pnas.1606320113. URL https://www.pnas.org/content/113/42/11776. Publisher: National Academy of Sciences Section: Physical Sciences.

[5] I. Barnes, K. H. Becker, and I. Patroescu. The tropospheric oxidation of dimethyl sulfide: A new source of carbonyl sulfide. *Geophysical Research Letters*, 21(22):2389–2392, 1994. ISSN 1944-8007. doi: 10.1029/94GL02499. URL https://onlinelibrary.wiley.com/doi/abs/10.1029/94GL02499. _eprint: https://onlinelibrary.wiley.com/doi/pdf/10.1029/94GL02499.

---

## Author Comment (AC1)

Response to Reviewers:

Thank you both for reviewing this manuscript. We appreciate the comments and questions raised that have helped improve the quality of this manuscript. Reviewer comments are reproduced below in blue text and author responses are in black text. For edits, additions to the text are in red and deletions are represented with a red strikethough. Line numbers referenced in the author responses reflect the line numbers in the condensed, tracked changes manuscript and SI.

Reviewer 1 Comments:

This study investigates the cloud processing of hydroperoxymethyl thioformate (HPMTF) in the marine boundary layer (MBL) using six-week measurements of cloud fraction and marine sulfur species, along with an observational constrained chemical box model. HPMTF is an intermediate oxidation product of dimethyl sulfide (DMS), the major sulfur species emitted from the ocean. By examining the loss of HPMTF to clouds, this study can support a more accurate quantification of carbonyl sulfide (OCS) and sulfur dioxide (SO2) formation and their contributions to new particle formation, and also improve parameterizations in global chemical transport models.

The manuscript is well written and accounts for a variety of sources that could cause uncertainness. However, there are two major concerns.

1. The measurements were conducted below a height of 10 m above ground level. A major concern is how well this can represent conditions near cloud levels. The manuscript indicates that this study is more relevant for well-mixed MBL conditions. Given the short lifetime of HPMTF, it would be helpful to perform a careful comparison of its lifetime and that of DMS with the MBL turnover time to understand the vertical profiles of these key sulfur species throughout the day. Specifically, the observation that the [DMS]/[HPMTF] ratio was low in cloud-free conditions but significantly higher below the cloud deck should be supported by an illustration of the vertical mixing dynamics of both DMS and HPMTF. This would help determine whether the observed changes are due to cloud processing of HPMTF or are instead a result of vertical mixing and dilution effects.

The reviewer raises an important point that we have thought about throughout this analysis. We agree that analyses conducted using ground-level measurements have their limitations and this is pervasive in our community as it is difficult (and often not possible) to assess how representative they are of the full boundary layer column. To more accurately make the comparison between [DMS]/[HPMTF] and cloud loss, vertical profiles of these species below, in, and above cloud are needed, which would come from an aircraft study. This has been done once prior to assess HPMTF cloud loss (Novak et al., 2021). However, aircraft campaigns are costly and can suffer from less sampling time (ex: a few flight legs), which can make it difficult to ascertain whether a few flight legs are representative of longer-term ambient conditions. While ground-based measurements cannot take the place of aircraft observations, there are some advantages (e.g., sampling statistics) and they help lay the foundation for future studies. The pros and cons of each method are discussed in detail in L351-L357 in Section 3.2.2.

We acknowledge the limitations of our method, particularly the near ground-level measurements, which was due to the short inlet required to minimize inlet loss of HPMTF and other oxidized molecules sampled by the Vocus AIM. We also evaluated the twice daily sonde vertical profiles to characterize the boundary layer for each study day (Fig. 5). This analysis of vertical profiles allows us to make an informed judgment on the extent to which our near ground-level measurements of the sulfur species were indicative of the sulfur species throughout the boundary layer, while still amassing sampling time during a ground study. We then reported cloud loss rates across all study days (includes days when ground-level measurements might not be indicative of cloud level measurements due to a stable boundary layer) and for just well-mixed days, where based on vertical profiles of [$H_2O$] and potential temperature, we expect our ground-level measurements of sulfur species to be consistent with their concentration at the cloud. We find that the major conclusion of fast loss of HPMTF to cloud holds in both scenarios.

We further consider how vertical mixing might have impacted our measurements and conclusions quantitatively. Our analysis indicates that the lifetime of HPMTF in the MBL is on average 0.5-1.9 hours, taking into account uncertainty in [HPMTF], and is dependent on the presence of cloud. Large eddy simulations suggest the timescale of MBL mixing is on the order of an hour (Feingold et al., 1998). Ideally, the HPMTF lifetime would be long relative to the MBL lifetime for surface measurements to be fully representative of the MBL column. However, we think that the near surface loss of HPMTF (surface deposition lifetime is 37 hours) is small relative to the MBL mixing time, thus damping this effect. The fact that the HPMTF lifetime is of order the MBL mixing time (only when clouds are present) is consistent with cloud loss being the dominant loss process. However, as we discuss above, the limitations of our measurements mean that some of the variability in [DMS]/[HPMTF], particularly the days with small residual loss terms, could also be due to MBL mixing. This work still highlights the importance of HPMTF cloud loss, based on a long time series of ground-level measurements, and allows for later aircraft studies (ex: AEROMMA) to provide information on cloud processing of HPMTF with less uncertainty.

In summary, we agree with you that ground-level measurements are a drawback of this analysis, but have presented the results fairly, while leaving room for future aircraft studies to build upon this work. We have updated the text in a few places to make these limitations and our reasons for them more clear.

L132: "The lower inlet height of the Vocus AIM relative to the RT-Vocus was a result of the Vocus AIM requiring a shorter inlet to minimize inlet loss of oxidized species."

L217: "Well-mixed boundary layers had vertical slopes in both potential temperature and water mixing ratio below the inversion layer, and were interpreted to mean that the concentration of sulfur species measured at the ground level represented their concentration at the cloud level."

L358: "It is our goal for this work to present longer-term measurements of DMS and HPMTF to inform our understanding of HPMTF cloud processing, and provide a basis for needed aircraft observations of vertically-resolved, well-calibrated HPMTF in this region in the future."

L573: "The fast loss of HPMTF to cloud should continue to be validated with future aircraft studies containing vertically-resolved measurements and coincident HPMTF calibration."

2. Another major concern is the method of calculating the cloud loss of HPMTF by comparing modeled results with observations. First, diurnally averaged measurements of meteorological factors and chemical species are used to constrain box modeling, which could lead to a significant bias, especially when comparing with measured HPMTF under varying atmospheric conditions. Although the study includes a simulation with time-varying temperatures, other key meteorological and chemical factors also vary and can significantly impact HPMTF concentrations. Second, as shown in Figure S1, the modeled results exceed the calibrated measurements of HPMTF, which indicates the calculation of differences will generate negative values to represent cloud loss thus doesn't make sense. The manuscript attributes this discrepancy to a potential 60% underestimation of measured HPMTF, which is not convincing and warrants further discussion. Also, it may be beneficial to incorporate a cloud term directly into the model box to facilitate a more accurate comparison and a clearer insight into the cloud processing of HPMTF.

The model does not include time-varying meteorological and chemical conditions to match each study day but rather models diurnal profiles of these conditions to develop a general method to assess cloud loss rates based on the ratio of a precursor and soluble, oxidized product. The decision to do this was based on the following reasons. The RT-Vocus (measuring DMS) and Vocus AIM (measuring HPMTF) had slightly different duty cycles to satisfy overall campaign priorities, and resulted in not continuous [DMS]/[HPMTF] coverage. This limits the ability to get an hourly, time-resolved cloud term from the model. Second, HPMTF was very sensitive to loss processes, resulting in many instances of near 0 [HPMTF] and very large [DMS]/[HPMTF], making assigning a time-varying loss term challenging to interpret. As a result, we focus on modelling diurnal profiles and comparing the window of stable afternoon [DMS]/[HPMTF]; the advantage of this method is the values are more interpretable, but comes with the drawback of not capturing loss due to variability in the cloud field (L421-L426). Furthermore, by focusing on comparing ratios of [DMS]/[HPMTF] rather than absolute concentrations of individual species in this method, we minimize the importance of the precursor [DMS] and thus needing a time-resolved term. Other chemical conditions that change in time and could have a larger impact on our analysis include [NO] and [RO$_2$], but we lack the measurements to constrain this.

The uncertainty in measured HPMTF, and how it impacts the retrieval of cloud loss terms based on comparing modeled and measured DMS/HPMTF, is another important concern in this analysis. First, we note that field measurements of molecules measured with Iodide CIMS can have large uncertainties. For a well-characterized Iodide CIMS instrument, this can be on the order of 30% (CIMS | NASA Airborne Science Program, 2024). Furthermore, a newly discovered molecule with no commercially available standard, such as HPMTF, can have uncertainties on the order of 20-50% (Jernigan et al., 2022; Novak et al., 2021; Vermeuel et al., 2020). Without generating HPMTF

in the lab, the comparison between modelled clear sky and measured clear sky provides the best opportunity to assess the combination of inlet loss of HPMTF and validity of the calibration factor, and thus quantify the uncertainty in our HPMTF measurement. The model for this comparison (shown in Fig. S1) was optimized with input conditions to match the clear sky day (temperature, relative humidity, gas-phase mixing ratios, solar zenith angle, etc.). This means that any uncertainty in the model is due to uncertainty in its loss terms ($k_{HPMTF+OH}$, aerosol uptake, deposition velocity). The up to 60% underestimation is the most accurate and fair way of reporting the data, and is reasonable given the challenges to making ambient field measurements of this particular molecule and in the context of the literature. We have explicitly noted the uncertainties in the text and reported ranges in all values derived from [HPMTF], such as cloud loss rates, to treat the uncertainty in the measurement fairly.

The cloud loss rate was determined by comparing measured [DMS]/[HPMTF] to modelled [DMS]/[HPMTF]. This comparison was completed for two values of measured [DMS]/[HPMTF] - one where HPMTF was calibrated by formic acid and one where HPMTF was "calibrated" by adjusting it with the 60% difference determined in Fig. S1. This results in larger HPMTF concentrations (Fig. S1), and results in two days out of thirty-one analyzed yielding negative cloud loss terms for the [DMS]/[HPMTF] comparison using the adjusted HPMTF concentration. This is due to the uncertainty in [HPMTF] (we report a *maximum* 60% difference) and uncertainty in the modelled loss processes (only a single measurement of aerosol uptake and deposition velocity exist) and inputs. However, for most days, the difference between clear sky [DMS]/[HPMTF] and modelled [DMS]/[HPMTF] is large, and beyond the uncertainty propagated from HPMTF concentrations. This suggests that the potential 60% underestimation, while not ideal, does not affect the conclusions of this paper that fast cloud processing of HPMTF is occurring and impacting $SO_2$ and OCS budgets. As discussed in the prior response, it is our goal that future work can build upon the findings in this paper and in particular, improve the findings by making vertically-resolved measurements, set up to reduce inlet loss, and calibrate for HPMTF.

Below are the relevant adjustments we have made to the text:

L346: "The reported range in cloud loss rates is determined based on uncertainty in the measured [HPMTF]. When using the model-derived HPMTF sensitivity, two of the 31 analysis days yielded a negative cloud loss term. In contrast, when using the formic acid derived sensitivity for HPMTF on cloud free days, we have an unaccounted residual loss term. For the majority of analyzed days, the difference between modelled and measured [DMS]/[HPMTF] is large enough that uncertainty in [HPMTF] does not impact the conclusions and reinforces fast cloud processing of HPMTF."

L573: "The fast loss of HPMTF to cloud should continue to be validated with future aircraft studies containing vertically-resolved measurements and coincident HPMTF calibration."

Reviewer 2 Comments:

Based on field measurements the authors present an assessment of how much the HPMTF loss to cloud affect the yields of SO2 and OCS from DMS oxidation. The manuscript is well written and conclusions sound. The analysis looks valid given the limited of knowledge about aqueous-phase oxidation of organo-sulfur compounds in the real atmosphere. I only have some remarks concerning how heterogeneous losses have been accounted for.

**Major comments**
**• Loss to aerosols**
In Eq. 1 the expression for the loss frequency, k, to aerosols is given in a specific form which might not be familiar to every reader. It took me a while to find the proper reference [1] and assure myself it is correct. Please add that reference. The equation for k is normally used with α instead of γ. For aerosol uptake the latter is more commonly used in an expression with only the second term as the cited DMS-relevant works [2, 3]. Why did the authors not used the usual form 1/4 v γ A? A short explanation would be help the reader.

We chose to show the specific form of the equation for aerosol loss in the paper to match how it is reported in the example for aerosol heterogeneous loss in Framework for 0-Dimensional Atmospheric Modeling v3.2, the framework with which we developed this analysis. We have now cited the correct reference and have provided a short explanation.

L189: "*... and uptake to marine aerosol particles was calculated according to Eq. 1* (Jacob, 2000), following the F0AM example for heterogeneous loss, where *A* is the aerosol surface area density, $D_g$ is the diffusivity in air, *r* is the aerosol radius, *v* is the mean molecular speed, and *γ* is the reactive uptake coefficient."

**• Loss to cloud droplets**
It has been reported that the DMS loss due to aqueous-phase oxidation by O3 is not negligible [4]. If considered by the authors, this would imply larger cloud loss of HPMTF in order to match the observed DMS/HPMTF ratios. Moreover, as it could be seen by the gamma values higher than for HPMTF, loss of DMSO, DMSO2, MSIA and MSA to clouds is likely affecting the SO2-yield as well. I understand the model used by the authors lacks explicit cloud chemistry but at least it would be good to have these points in the discussion of results.

Thanks for bringing up this important point. You are correct, this is not included in the model we developed, but we have now included some discussion of this point in the main text. We have combined the response below with a response from the next comment as well.

L449: "It is also possible that SO$_2$ yields from DMS oxidation in this model could be further reduced by including the isomerization of CH$_3$SOO to CH$_3$SO$_2$ (Chen et al., 2023) (yielding a

reduction in clear sky diurnally-averaged $SO_2$ from DMS by 6%), and by including aqueous-phase oxidation of DMS by $O_3$. Recent research has shown the oxidation of DMS by $O_3$ can be significant, forming DMSO, MSIA, and MSA (Hoffmann et al., 2016), all molecules with high reactive uptake coefficients (Table S1), which lead to reductions in DMS-derived $SO_2$."

**Minor comments**
• p. 3, l. 75: Upon O2 addition to the CH3S radical CH3SOO is produced and not CH3SO2. The latter comes later from CH3SOO isomerization and the T-dependent branching ratio between channels yielding either SO2 or MSA are for reactions involving CH3SO2. However, the CH3SOO isomerization is not included neither in MCM nor in the updated F0AM model as one can learn from Table S1. The author should consider how much of an impact it has on the overall SO2 conversion from DMS oxidation in their model.

Thanks for pointing out this omission. The MCM has $O_2$ addition to the CH3S radical forming CH3SOO. However, we did not have the isomerization of CH3SOO forming CH3SO2. We have tested adding this to the mechanism in the model, with the rate constant 7e14*exp(-9659/T) (Chen et al., 2023). Including the isomerization of CH3SOO to CH3SO2 reduces the yield of $SO_2$ from DMS (calculated as the 24-hour average of PSO$_2$ / LDMS) by 6% in the clear sky and the impact increases at higher cloud loss rates.

L76: "The OH-oxidation of MeSH and subsequent $O_2$ addition forms the $CH_3SO_2O$ radical; the $CH_3SOO$ radical isomerizes to $CH_3SO_2$, which has a temperature-dependent branching ratio forming $SO_2$ or MSA (Chen et al., 2023)."

L449: "It is also possible that $SO_2$ yields from DMS oxidation in this model could be further reduced by including the isomerization of $CH_3SOO$ to $CH_3SO_2$ (Chen et al., 2023) (yielding a reduction in clear sky diurnally-averaged $SO_2$ from DMS by 6%), and by including aqueous-phase oxidation of DMS by $O_3$. Recent research has shown the oxidation of DMS by $O_3$ can be significant, forming DMSO, MSIA, and MSA (Hoffmann et al., 2016), all molecules with high reactive uptake coefficients (Table S1), which lead to reductions in DMS-derived $SO_2$."

• p. 18, l. 448: Saying that the results of this work "align" with the OCS-yield determined in the experiments by [5] is misleading. Those experiments were done in a chamber under nonatmospheric conditions and surely without liquid water. I would rather say it is a fortuitous coincidence that the two OCS-yields meant here are similar.

Agreed, this was a misleading choice of words. We have adjusted the text as follows:

L476: "The historical model of OCS formation from DMS OH-oxidation also uses a direct, fixed yield of 0.007 (Barnes et al., 1994), which coincidentally, is similar to the  findings here in the presence of cloud."

We have also made some additions to the text to better represent the coarse mode aerosol surface area during the study.

L369: "Additionally, while we do not have concurrent measurements of coarse mode sea spray aerosol particles, we have estimated the wet surface area of particles with dry diameters larger than 0.47 µm using scattering coefficients measured by an integrated Nephelometer. The relationship between dry surface area and scattering coefficient used for this estimation was derived from measurements collected during the Aerosol and Cloud Experiments in the Eastern North Atlantic (ACE-ENA) campaign (Wang et al., 2022), where an Aerodynamic Particle Sizer (APS) provided direct measurements of coarse mode dry surface area. The wet surface area was then calculated based on the hygroscopic growth factor of sea-salt aerosols as a function of relative humidity (Pitchford et al., 2007). The average and interquartile range ($D_p > 0.47$ µm) was 30 (15-61) $\mu m^2$ $cm^{-3}$, though many higher instances occurred. Sea spray aerosol particles  are hygroscopic (Zieger et al., 2017) and  provide an enhanced surface area for HPMTF uptake, particularly during strong winds that promote sea spray production, that is not captured in the model. Based on ACE-ENA observations showing a minimum cloud droplet number concentration on the order of 30 $cm^{-3}$ and an average droplet size of 10 µm (Wang et al., 2022), the cloud droplet surface area would be 9400 $\mu m^2$ $cm^{-3}$, at least 150 times that of the aerosol surface area. As a result, HPMTF loss to aerosol  is still expected to be  lower than cloud due to the large difference in surface area between aerosol and cloud droplets."

References
CIMS | NASA Airborne Science Program: https://airbornescience.nasa.gov/category/type/CIMS, last access: 6 October 2024.

Barnes, I., Becker, K. H., and Patroescu, I.: The tropospheric oxidation of dimethyl sulfide: A new source of carbonyl sulfide, Geophys. Res. Lett., 21, 2389–2392, https://doi.org/10.1029/94GL02499, 1994.

Chen, J., Lane, J. R., Bates, K. H., and Kjaergaard, H. G.: Atmospheric Gas-Phase Formation of Methanesulfonic Acid, Environ. Sci. Technol., 57, 21168–21177, https://doi.org/10.1021/acs.est.3c07120, 2023.

Feingold, G., Kreidenweis, S. M., & Zhang, Y: Stratocumulus processing of gases and cloud condensation nuclei: 1. Trajectory ensemble model, Journal of Geophysical Research, 103(D16), 19,527–19,542, https://doi.org/10.1029/98JD01750, 1998.

Hoffmann, E. H., Tilgner, A., Schrödner, R., Bräuer, P., Wolke, R., and Herrmann, H.: An advanced modeling study on the impacts and atmospheric implications of multiphase dimethyl sulfide chemistry, Proc. Natl. Acad. Sci., 113, 11776–11781, https://doi.org/10.1073/pnas.1606320113, 2016.

Holmes, C. D., Bertram, T. H., Confer, K. L., Graham, K. A., Ronan, A. C., Wirks, C. K., and Shah, V.: The Role of Clouds in the Tropospheric NOx Cycle: A New Modeling Approach for Cloud Chemistry and Its Global Implications, Geophysical Research Letters, 46, 4980–4990, https://doi.org/10.1029/2019GL081990, 2019.

Jacob, D.: Heterogeneous chemistry and tropospheric ozone, Atmos. Environ., 34, 2131–2159, https://doi.org/10.1016/S1352-2310(99)00462-8, 2000.

Jernigan, C. M., Fite, C. H., Vereecken, L., Berkelhammer, M. B., Rollins, A. W., Rickly, P. S., Novelli, A., Taraborrelli, D., Holmes, C. D., and Bertram, T. H.: Efficient Production of Carbonyl Sulfide in the Low-NOx Oxidation of Dimethyl Sulfide, Geophys. Res. Lett., 49, e2021GL096838, https://doi.org/10.1029/2021GL096838, 2022.

Novak, G. A., Fite, C. H., Holmes, C. D., Veres, P. R., Neuman, J. A., Faloona, I., Thornton, J. A., Wolfe, G. M., Vermeuel, M. P., Jernigan, C. M., Peischl, J., Ryerson, T. B., Thompson, C. R., Bourgeois, I., Warneke, C., Gkatzelis, G. I., Coggon, M. M., Sekimoto, K., Bui, T. P., Dean-Day, J., Diskin, G. S., DiGangi, J. P., Nowak, J. B., Moore, R. H., Wiggins, E. B., Winstead, E. L., Robinson, C., Thornhill, K. L., Sanchez, K. J., Hall, S. R., Ullmann, K., Dollner, M., Weinzierl, B., Blake, D. R., and Bertram, T. H.: Rapid cloud removal of dimethyl sulfide oxidation products limits SO2 and cloud condensation nuclei production in the marine atmosphere, Proc. Natl. Acad. Sci., 118, e2110472118, https://doi.org/10.1073/pnas.2110472118, 2021.

Pitchford, M., Malm, W., Schichtel, B., Kumar, N., Lowenthal, D., Hand, J.: Revised Algorithm for Estimating Light Extinction from IMPROVE Particle Speciation Data, J. Air Waste Manage. Assoc., 57, 1326-1336, https://doi.org/10.3155/1047-3289.57.11.1326, 2007.

Vermeuel, M. P., Novak, G. A., Jernigan, C. M., and Bertram, T. H.: Diel Profile of Hydroperoxymethyl Thioformate: Evidence for Surface Deposition and Multiphase Chemistry, Environ. Sci. Technol., 54, 12521–12529, https://doi.org/10.1021/acs.est.0c04323, 2020.

Wang, J., et al.: Aerosol and Cloud Experiments in the Eastern North Atlantic (ACE-ENA), Bull. Am. Meteorol. Soc., 103, E619-E641, https://doi.org/10.1175/bams-d-19-0220.1, 2022.

---

## Author Response (AR2)

Editor's Comments:

I hereby accept your manuscript describing the use of ground based observations to study the cloud processing of DMS oxidation products for publication in ACP. There is just one technical point I noticed in line 76, where you write CH3SOO2 as the first intermediate in the oxidation of MeSH. Is this correct, or should it be CH3SOO? I suggest you carefully check the chemical formulae in this sentence when preparing your files for publication.

Author's Response:

Thank you for the support on this manuscript and for the careful review. We have checked the version of the manuscript we submitted following reviewer's comments and it does not mention CH3SOO2. We write that CH3SOO is the first intermediate of MeSH, as you indicated as well. We have also confirmed this chemistry is correct based on the referenced material.

Below is the relevant text:

"The OH-oxidation of MeSH and subsequent $O_2$ addition forms the $CH_3SOO$ radical; the $CH_3SOO$ radical isomerizes to $CH_3SO_2$, which has a temperature-dependent branching ratio forming $SO_2$ or MSA (Chen et al., 2023). Recent computational work has shown the $SO_2$ yield from $CH_3SO_2$ is 99% at 300 K, but drops to 4% at 260 K (Chen et al., 2023)."